# Essential Oils as Antioxidants: Mechanistic Insights from Radical Scavenging to Redox Signaling

**DOI:** 10.3390/antiox15010037

**Published:** 2025-12-26

**Authors:** Yeqin Huang, Haniyeh Ebrahimi, Elena Berselli, Mario C. Foti, Riccardo Amorati

**Affiliations:** 1Department of Chemistry “G. Ciamician”, University of Bologna, Via Gobetti 83, 40129 Bologna, Italy; yeqin.huang2@unibo.it (Y.H.);; 2Consiglio Nazionale delle Ricerche, Istituto di Chimica Biomolecolare, Via Paolo Gaifami 18, 95126 Catania, Italy

**Keywords:** essential oils, antioxidant mechanisms, methodology, kinetics, radicals, phenolic compounds, Nrf2/Keap1 pathway, redox signaling, terpenes

## Abstract

Essential oils (EOs) are complex volatile mixtures that exhibit antioxidant activity through both chemical and biological pathways. Phenolic constituents act as efficient chain-breaking radical-trapping antioxidants, whereas some non-phenolic terpenes operate through distinct mechanisms. Notably, γ-terpinene functions via a “radical export” pathway, generating hydroperoxyl radicals that intercept lipid peroxyl radicals and accelerate chain termination. Recent methodological advances, such as inhibited autoxidation kinetics, oxygen-consumption assays, and fluorescence-based lipid peroxidation probes, have enabled more quantitative evaluation of these activities. Beyond direct radical chemistry, EOs also regulate redox homeostasis by modulating signaling networks such as Nrf2/Keap1, thereby activating antioxidant response element–driven enzymatic defenses in cell and animal models. Phenolic constituents and electrophilic compounds bearing an α,β-unsaturated carbonyl structure may directly activate Nrf2 by modifying Keap1 cysteine residues, whereas non-phenolic terpenes likely depend on oxidative metabolism to form active electrophilic species. Despite broad evidence of antioxidant efficacy, molecular characterization of EO–protein interactions remains limited. This review integrates radical-chain dynamics with redox signaling biology to clarify the mechanistic basis of EO antioxidant activity and to provide a framework for future research.

## 1. Introduction

Essential oils (EOs) are volatile and aromatic secondary metabolites found in various plant organs, such as flowers, leaves, stems, bark, and roots [1]. They are complex mixtures of terpenes, terpenoids, and a variety of minor aromatic or aliphatic compounds [2]. The composition of EOs is determined by both genetic and environmental factors. These factors include plant species, specific part of the plant used, geographical origin, climate, and cultivation conditions [3]. Extraction methods can influence the chemical profile and bioactivity of essential oils, including hydro-distillation, steam distillation, cold pressing, solvent extraction and supercritical fluid extraction [4]. EOs have been traditionally used for their aroma and related functional properties, and are also applied as natural flavorings and preservatives in foods and beverages [5,6].

In recent years, interest in EOs has considerably grown because of their biological activities, among which antioxidant properties attracted increased recognition in food science and pharmacology (Figure 1). Their major constituents—terpenes, terpenoids and phenylpropanoids—exhibit characteristic redox behavior and support applications in food preservation, packaging, skincare, and nutraceuticals. Traditionally, the antioxidant properties of EOs have been attributed to their phenolic components [7]. Phenols are known for their ability to scavenge free radicals and chelate pro-oxidant metal ions [8]. Recent studies have shown that some non-phenolic compounds, in particular those having a pro-aromatic 1,4-cyclohexadiene structural motif, can also play a significant role in the antioxidant activity of EOs [9,10]. These compounds help interrupt oxidative chain reactions, thereby offering protection to both food products and living organisms from oxidative stress [11]. With growing understanding of their antioxidant mechanisms, the use of EOs as natural antioxidants has attracted considerable attention in food processing, cosmetics, and other fields. In food applications, EOs are used to slow down lipid oxidation and maintain product quality [12]. In cosmetics, EOs are added to formulations to mitigate the oxidative stress linked to skin aging and inflammation [13]. By limiting oxidative damage, EOs may also contribute to the prevention and management of chronic diseases such as cardiovascular and neurodegenerative disorders [14]. Concerns regarding the safety of synthetic antioxidants have further intensified interest in effective natural alternatives [15]. Owing to their intrinsic antioxidant properties, EOs are now considered promising agents for natural preservation and health-related applications.

A reliable assessment of the antioxidant activity is fundamental for both scientific research and practical applications, as it enables the identification and comparison of effective antioxidant agents. Preliminary studies on the antioxidant activity were based on simplified chemical assays such as 2,2-diphenyl-1-picrylhydrazyl radical scavenging assay (DPPH), 2,2′-azino-bis(3-ethylbenzothiazoline-6-sulfonic acid) radical cation decolorization assay (ABTS), ferric reducing antioxidant power assay (FRAP) and oxygen radical absorbance capacity assay (ORAC), which offered convenience and a rapid screening [16]. However, these tests are often unable to capture the complexity of the antioxidant mechanisms in real food matrices and living organisms [17]. More recent research has explored advanced chemical methods, cell-based models, and, to a lesser extent, in vivo approaches, aiming for a greater physiological relevance [18,19]. Despite these advances, large margins of uncertainty still exist, including the lack of standardization, the inherent variability in EO composition, and finally, difficulties in comparing the results of different studies. A comprehensive review in this context is, therefore, not only timely but also necessary to identify gaps, improve the strategies of evaluation and guide the rational use of EOs as natural antioxidants in food technology, cosmetic formulations, and medical practice.

This review examines the antioxidant activity of EOs from a mechanistic perspective, with the aim of bridging radical-chain chemistry and redox signaling regulation. Emphasis is placed on both classical chain-breaking radical-trapping mechanisms and non-classical pathways of radical interception. Current quantitative strategies, including inhibited autoxidation kinetics, oxygen-consumption measurements, and fluorescence-based probes, are critically evaluated in terms of their mechanistic resolution and inherent limitations, alongside complementary in vitro and in vivo models. Emerging methodologies with potential to advance EO antioxidant research are also highlighted. Finally, recent progress in understanding the modulation of the Nrf2/Keap1 pathway by EO constituents is discussed, with particular attention to the contributions of phenolic antioxidants and electrophilic non-phenolic compounds.

## 2. Methodology

The literature was surveyed using the Web of Science, Google Scholar, and PubMed databases. For antioxidant mechanisms and methodological aspects, searches were conducted using combinations of keywords related to radical propagation, lipid peroxidation, antioxidant assays, and methodology, without restriction on publication year, while prioritizing key studies from the past decade to capture recent conceptual and methodological advances. Studies addressing the antioxidant activity of EOs and their constituents were identified using keywords related to essential oils, terpenes, oxidative stress, Nrf2/Keap1 signaling, and ferroptosis. In this case, the search was limited to articles published between 2000 and September 2025. Selection was based on relevance to lipid peroxidation, cellular redox regulation, and mechanistic or cell-based evidence.

## 3. Mechanisms of Lipid Peroxidation

To evaluate the antioxidant potential of EOs, it is crucial to clearly define their primary target: the inhibition of lipid peroxidation in materials and the mitigation of oxidative stress in biological systems. Polyunsaturated fatty acids (PUFAs) are especially susceptible to peroxidation, both in stored materials and in living organisms, as they are abundant in cellular membranes. In foods, cellular membranes, and low-density lipoproteins (LDLs), PUFAs exposed to oxygen readily undergo an irreversible process called peroxidation or autoxidation. This reaction alters the flavor and aroma of edible products and generates toxic metabolites that can ultimately lead to cell death. Lipid peroxidation is among the most widespread organic reactions. It can be delayed or prevented through the use of antioxidants [20]. Despite the large variety of systems in which peroxidation occurs, the underlying mechanism is surprisingly similar. It involves a radical-chain reaction propagated by oxygen-centered peroxyl radicals (ROO^•^) [21]. Due to the extremely rapid reaction of carbon-centered (alkyl) radicals (R^•^) with molecular oxygen (triplet O_2_), virtually all R^•^ species are converted to ROO^•^ even at low O_2_ concentrations, making ROO^•^ the predominant radical species in lipid peroxidation. Lipid peroxidation can broadly be divided into three mechanistic phases, illustrated in Figure 1A.

### 3.1. Initiation

Initiation involves the formation of radicals within the system. In food and other lipid-rich materials, this can occur via several pathways, including: a direct reaction of O_2_ with bis-allylic C–H bonds (often accelerated by heat) [22], photosensitized reactions [23], or metal-catalyzed decomposition of pre-existing hydroperoxides [24]. Due to the chain-propagating nature of the process, even a minimal rate of radical generation can result in significant oxidative damage over time, as a single initiating radical can lead to the oxidation of numerous lipid molecules.

### 3.2. Propagation

Propagation entails the reaction of the ROO^•^ radical with a lipid substrate, leading to the formation of a lipid hydroperoxide (ROOH) and a new carbon-centered radical. This radical quickly reacts with O_2_ to regenerate ROO^•^, perpetuating the chain. In lipid systems, propagation predominantly affects PUFAs, as their bis-allylic hydrogens are several hundred times more reactive (at room temperature) towards ROO^•^ than allylic or aliphatic sites [21]. However, the mechanisms of propagation can be more complex, potentially involving ROO^•^ fragmentation into species such as hydroperoxyl radicals (HOO^•^) or alkoxyl radicals (RO^•^), thereby generating stable oxygen products like epoxides or carbonyl compounds [25].

### 3.3. Termination

Termination involves the removal of radical species from the system, predominantly through bimolecular reactions between radicals. For instance, two ROO^•^ radicals may recombine to form a carbonyl compound and an alcohol. Although radical–radical reactions are intrinsically fast, the very low steady-state concentration of radicals in lipid systems makes termination a statistically rare process. This rarity becomes even more pronounced in compartmentalized systems, such as emulsions, where restricted diffusion further reduces the likelihood of productive radical encounters [21].

### 3.4. Products of Lipid Peroxidation

The products of lipid peroxidation can be broadly classified into early-stage products, generated directly during the radical chain process, and late-stage products, which arise from the subsequent degradation or transformation of these initial species. However, this distinction is not absolute, as certain compounds can originate from both direct radical pathways and secondary transformation processes (see, for example, Figure 1B).

#### 3.4.1. Early Products

Early-stage products are predominantly hydroperoxides, especially in the case of PUFAs. For linoleate residues, the complete product distribution is well characterized (see Figure 2). The double bond geometry (cis/trans) of the resulting hydroperoxides depends on how rapidly peroxyl radicals are intercepted by H-atom donors [26]. Consequently, during the autoxidation of triacylglycerols, trans–trans hydroperoxides typically dominate over cis–trans isomers. Interestingly, this ratio inverts in the presence of antioxidants, which favor the formation of cis–trans species by accelerating radical quenching. Hydroperoxides derived from polyunsaturated lipids also feature conjugated double bonds, enabling straightforward detection by UV–vis spectrophotometry [27]. The product distribution becomes more complex as the number of double bonds in the fatty acids increases. This is due to the fast cyclization reactions that may involve both peroxyl and alkyl radicals. In the case of substrates with one unsaturation, such as oleic acid and cholesterol, the H-atom abstraction by ROO^•^ which yields hydroperoxides is in competition with ROO^•^ addition to the double bond. Radical addition on non-conjugated double bonds typically generates epoxides and secondary alkoxyl radicals, RO^•^ (see Figure 1), which can rearrange to α-hydroxyl alkyl radicals. These latter, upon reaction with O_2_, yield ketones and HOO^•^ radicals [28].

#### 3.4.2. Late Products

Late product formation from unstable hydroperoxides is promoted by homolytic or heterolytic reactions. The concentration of hydroperoxides increases at the beginning of the lipid peroxidation only. Afterwards, their concentration decreases because they convert into more stable secondary products, which accumulate over time [30]. In this regard, the oxidative stability of lipid materials is usually evaluated by measuring the content of both hydroperoxides and late products (typically volatile and non-volatile aldehydes) [31]. Although the reaction network leading to secondary products is very complex (for a comprehensive review see refs. [21,26]), two main pathways of hydroperoxide decay can be recognized: Hock fragmentation and radical homolysis. The latter decomposition process, triggered by reduced transition metals (such as Fe^2+^) or elevated temperatures, leads to the formation of alkoxyl radicals. The reaction mechanism may proceed via the generation of α-hydroxyalkyl radicals or, alternatively, through β-scission, producing alkyl radicals and aldehydes (see Figure 3). This pathway is highly relevant for the initiation phase of lipid peroxidation, as trace amounts of both metals and hydroperoxides are invariably present in lipid-containing systems [32]. In addition to the above reactions, there is another process known as “Hock fragmentation” which is catalyzed by acids. This process, through a non-radical pathway, transforms a hydroperoxide into two aldehydes and is fundamental for the formation of oxylipins from cholesterol [21]. The nature of the aldehydes formed during lipid peroxidation depends critically upon the fatty acid being oxidized. It may include mono- and di-carbonyls with various degrees of unsaturation (like malondialdehyde or 4-hydroxynonenal, see Figure 3). As expected, ω-3 fatty acids give aldehydes with a shorter chain (and a higher volatility) than ω-6 [33]. Hydroperoxides can also react with alkenes, forming epoxides, which in turn can be attacked by water or acids, forming diols or diesters, respectively [34]. Besides carbonyls and epoxides, other minor products that derive from advanced oxidation of aldehydes are also formed, like furans, aldol condensation products [33], and short-chain acids like acetic and formic acid, which are the main water-soluble volatiles that are detected by the Rancimat assay [35].

## 4. Mechanistic Basis of the Antioxidant Activity

Antioxidants are molecules—or enzymatic systems—that, even at relatively low concentrations, can slow down or halt the peroxidation of an oxidizable substrate. Owing to their chemical diversity and distinct modes of action, they are commonly classified according to the stage of the peroxidation process at which they intervene.

### 4.1. Preventive Mechanisms

Preventive antioxidants act at the initiation stage of lipid peroxidation by suppressing radical formation. In most systems, initiation is dominated by Fenton-type reactions, where reduced transition metal ions (Fe^2+^, Cu^+^) catalyze the homolytic cleavage of hydroperoxides, generating alkoxyl radicals. Consequently, the two main preventive strategies are peroxide decomposition and metal ion chelation. In biological systems, glutathione peroxidase (GPX) and catalase are the key enzymes responsible for peroxide reduction. In contrast, non-biological matrices contain very few examples of catalytic peroxide decomposition, mostly involving sulfur- or selenium-based systems [37]. Metal chelators, whether natural (e.g., phytic acid) or synthetic (e.g., ethylenediaminetetraacetic acid, EDTA), exert their protective effect by stabilizing metal ions in a single redox state, thereby preventing redox cycling. In emulsified systems, chelators also operate by physically isolating metal ions from hydroperoxides, hindering catalytic contact. Effective chelating agents typically contain multiple polar or ionizable functional groups—such as phenoxides, carbonyls, carboxylates, or amines. Since EOs lack such functionalities, they are unlikely to display significant preventive activity through metal chelation.

### 4.2. Chain-Breaking

Chain-breaking antioxidants, also referred to as radical-trapping antioxidants, function by quenching lipid ROO^•^ (see Figure 4). Most low-molecular-weight natural antioxidants, such as tocopherols, phenolic acids, and flavonoids, belong to this category, along with common synthetic additives like tert-butylhydroquinone (TBHQ), BHT, and propyl gallate. These compounds are consumed stoichiometrically during inhibition, meaning their protective effect is inherently limited by their initial concentration. Their efficiency can be significantly enhanced through regeneration mechanisms, particularly in biological membranes, where lipophilic antioxidants, such as α-tocopherol, localize within the hydrophobic core. There, they can be regenerated by hydrophilic reducing agents—most notably ascorbate, which is present in the cytosol at concentrations approaching 1 mM—allowing sustained antioxidant protection beyond their initial supply.[20] Phenolic antioxidants, along with other radical-trapping antioxidants, inhibit lipid peroxidation by donating a hydrogen atom to the chain-carrying lipid ROO^•^, according to the reaction:AH + ROO^•^ → A^•^ + ROOH

The efficiency of this reaction is expressed by its rate constant (*k*_inh_), which in apolar environments—such as biological membranes—can reach values as high as 10^4^–10^7^ M^−1^ s^−1^. For uninhibited peroxidation, the overall rate is described by:−d[O_2_]/dt = *k*_p_ × *R*_i_^1/2^[RH]/(2k_t_)^1/2^ where *k_p_* is the propagation rate constant, 2*k*_t_ the termination rate constant, R_i_ the initiation rate, and finally [RH] the concentration of the substrate. In the presence of a chain-breaking inhibitor, the oxidation rate becomes:−d[O_2_]/dt = *k*_p_ × *R*_i_ × [RH]/(*n k*_inh_ [inhibitor]) where *n* is the stoichiometric factor, representing the number of ROO^•^ radicals trapped per antioxidant molecule:*n*ROO^•^ + Inhibitor → products

To summarize, the key requirements for an effective chain-breaking antioxidant (AH) are [38]:(1)Non-toxicity.(2)High reactivity toward lipid peroxyl radicals, i.e., a large inhibition rate constant (*k_inh_*).(3)Stability in air, under the reaction conditions, is extremely important because if AH reacted with O_2_ to give A^•^ and HOO^•^ (AH + O_2_ → A^•^ + HOO^•^), it would behave as a pro-oxidant rather than an antioxidant.(4)Formation of a stable radical A^•^ in the reaction AH + ROO^•^ → A^•^ + ROOH does not propagate the oxidative chain (A^•^ + RH →AH + R^•^).(5)No reaction of A^•^ with O_2_ (A^•^ + O_2_ → AOO^•^), as this reaction would also propagate the chain reaction.

Molecular scaffolds satisfying most of these criteria include phenols, as well as certain enols (e.g., ascorbate, Maillard reaction products) [39] and some amines [40]. On the other hand, Rule 5 generally excludes that hydrocarbons can function as antioxidants since their carbon-centered radicals almost invariably react with O_2_ to give peroxyl radicals, see Figure 1 and Figure 3. However, notable exceptions exist—some of which are particularly relevant to the chemistry of EOs and will be discussed below.

#### 4.2.1. Phenolic Antioxidants

Phenols are among the most potent natural antioxidants. Their ability to slow down lipid peroxidation is due to the transfer of a hydrogen atom from the phenolic hydroxyl group (ArO–H) to a lipid ROO^•^ through a formal hydrogen atom transfer (HAT) (Figure 5).

The rate constant for this process, *k*_inh_, is determined by enthalpic factors (e.g., the bond dissociation enthalpy of the O–H bond), entropic factors (e.g., steric hindrance around the hydroxyl group), and solvent effects. In particular, hydrogen bonding between phenolic OH and solvent can markedly reduce the reactivity of phenols toward ROO^•^. In apolar media such as hydrocarbon solvents, phenolic antioxidants can display *k*_inh_ values as high as 10^4^–10^7^ M^−1^ s^−1^ [41,42]. In contrast, in polar solvents and in water, the formation of an H-bond ArO-H···OH_2_ drastically reduces *k*_inh_ as shown in Figure 5C. This reduction is also caused by polar functional groups such as ester moieties in triglycerides or phosphate headgroups in phospholipids. Generally, the higher the polarity of the solvent or of the functional group, the lower the rate constant for HAT.

The fate of the phenoxyl radical is crucial (see Rule 4) in determining the overall efficiency of antioxidant protection. Simple monophenols are able to trap two peroxyl radicals (stoichiometric coefficient *n* = 2), unless radical–radical recombination occurs, in which case only one ROO^•^ is neutralized per phenol (Figure 5A). In contrast, hydroquinones bearing electron-donating substituents often display lower stoichiometric coefficients. This reduction arises because their semiquinone intermediates can react with O_2_, generating HOO^•^, which in turn may sustain the oxidative chain (Figure 5B).

Many natural lipids contain significant quantities of phenolic antioxidants. For instance, seed oils typically have high levels of tocopherols or tocotrienols, whereas olive oil contains polyphenols such as oleuropein and hydroxytyrosol. Among EO antioxidant components, phenols such as carvacrol, thymol, eugenol, guaiacol and methylphenols (Figure 6) behave as radical trapping antioxidants, although with a much smaller *k*_inh_ than α-tocopherol [43].

#### 4.2.2. γ-Terpinene as Antioxidant

While the antioxidant properties of phenols are well established, the ability of certain non-phenolic molecules belonging to the terpene family and able to retard lipid peroxidation has received much less attention. A notable example is γ-terpinene, which, despite being highly oxidizable and requiring careful storage, exhibits significant antioxidant activity. Its mode of action has been described as “termination enhancement”, owing to its ability to accelerate the termination of peroxyl radicals, see Figure 1.

Under normal conditions, ROO^•^ radicals decay via the self-reaction:ROO• + ROO• → ROOOOR which, for secondary or primary alkyl peroxyls (R_2_CHOO^•^ or RCH_2_OO^•^), decomposes to yield singlet oxygen (^1^O_2_) and disproportionation products. Termination can be accelerated by the hydroperoxyl radicals (HOO^•^) which leads to the following cross-reaction:HOO• + ROO• → ROOH + O_2_

Generally, this reaction proceeds faster than ROO^•^ self-reaction. This cross-termination lowers the steady-state concentration of ROO^•^ and thereby decreases the rate of peroxidation (–d[O_2_]/dt).

γ-Terpinene reacts with lipid peroxyl radicals (ROO^•^) to generate an unstable γ-terpinyl peroxyl intermediate, which rapidly decomposes to yield hydroperoxyl radicals (HOO^•^) and p-cymene. Although HOO^•^ is, in principle, capable of propagating lipid peroxidation, in this system its predominant role is to promote ROO^•^–ROO^•^ termination, thereby reducing the overall rate of substrate (RH) oxidation. This mechanism accounts for the marked retardation of linoleic acid peroxidation observed when γ-terpinene is present at millimolar concentrations in organic solution [44,45,46].

A similar effect is also observed in the oxidation of slow-reacting substrates such as isopropylbenzene (cumene), which forms tertiary ROO^•^ radicals characterized by small self-termination rate constants (*k*_t_ ≈ 10^4^ M^−1^ s^−1^). In such cases, the peroxidation process is slowed down by EO components that generate secondary ROO^•^ radicals—such as linalool, limonene, or citral—which have significantly higher termination rate constants [47].

γ-Terpinene has recently been shown to suppress lipid peroxidation in micelles and liposomes through a novel “radical-exporting” mechanism. In these biphasic environments, PUFA oxidation occurs within the lipophilic core of micelles or lipid bilayers, where lipid peroxyl radicals (ROO^•^) propagate the reaction efficiently. Upon addition of γ-terpinene, ROO^•^ radicals in the lipid phase are rapidly quenched, generating HOO^•^ radicals (as discussed above). Due to their small size and high polarity, HOO^•^ radicals diffuse out of the lipid compartment into the aqueous phase, thereby interrupting radical propagation within the lipid domain. This process has been described as a “slingshot” or “radical-exporting” mechanism, as the newly formed HOO^•^ radicals are effectively expelled from the lipid environment, preventing further chain propagation where it is most efficient (Figure 2) [10].

## 5. Chemical Methods

Inhibited autoxidation methods are considered the gold standard for assessing antioxidant activity, as they measure a compound’s ability to protect a substrate from oxidation. The rate of autoxidation is monitored in the presence or absence of antioxidants using parameters such as oxygen consumption (oximetry), hydroperoxide formation, or substrate depletion. When EOs are investigated using kinetic assays, the intrinsic instability of their constituents must be considered, as storage-induced autoxidation and secondary oxygenation can alter their chemical composition [48]. Monoterpene hydrocarbons, for instance, are particularly prone to forming hydroperoxides and other oxidized derivatives, which often display reactivities distinct from those of the parent compounds [2]. Such transformations can bias the apparent inhibitory efficiency determined in autoxidation assays and complicate the interpretation of antioxidant mechanisms. Periodic verification of EO composition prior to kinetic evaluation is therefore essential to ensure that the measured activity reflects the native chemical profile rather than storage-related artifacts.

### 5.1. Oximetry

Oximetry offers a direct and continuous measure of peroxidation by tracking O_2_ consumption, typically inferred from the pressure drop in a closed vessel. It is essential to ensure that O_2_ is the only reactive gas during the experiment [43]. Direct measurement can also be achieved using a Clark electrode [49] or an optical sensor based on fluorescence quenching, which is particularly suitable for lipids dispersed in water [10,22]. A representative oximetry trace is shown in Figure 3, illustrating the inhibition period induced by antioxidants. The primary advantage of this technique is that it monitors peroxidation kinetics in real-time without the need for sampling, providing a clear and continuous assessment of antioxidant efficacy.

### 5.2. Hydroperoxides

Hydroperoxides (ROOH) formation is a commonly used parameter to monitor lipid peroxidation, as ROOH are the primary products of radical attack on unsaturated lipids. They can be detected through their redox reactivity, for example, using the Fe^2+^–thiocyanate assay, which relies on the oxidation of Fe^2+^ by hydroperoxides. Alternatively, hydroperoxides can be identified via their characteristic spectroscopic signatures, such as FT-IR absorbance of the OO-H stretch or ^1^H-NMR signals of the hydroperoxide proton. A popular way to measure the formation of hydroperoxides in samples containing polyunsaturated lipids is to determine conjugated dienes (CD). For example, linoleic acid contains two non-conjugated double bonds at positions 9 and 12. The peroxidation of this substrate, as for other PUFAs with multiple non-conjugated double bonds, produces CD hydroperoxides as primary oxidation products, see Figure 1 [51]. These compounds have a strong absorption band at 234 nm with large molar extinction coefficients that are slightly dependent on the solvent, as the values are 25,500 M^−1^cm^−1^ in cyclohexane, 29,100 M^−1^cm^−1^ in acetonitrile and 30,500 M^−1^cm^−1^ in *tert*-butanol [52]. The extent of oxidation can be evaluated by following the formation of CD over time, *V*_CD_ = d*A*_234_/ed*t*, where *V*_CD_ is the rate of CD formation, *A*_234_ the absorbance at 234 nm and **e** the extinction coefficients at 234 nm.

Other techniques used to study inhibited lipid peroxidation include isothermal calorimetry [53,54], which measures the heat released during the propagation phase, and the quantification of volatile products deriving from hydroperoxide decomposition such as hexanal using GC–MS [55,56]. Recent developments aimed at increasing throughput involve the use of oxidizable fluorescent probes, which co-oxidize with the substrate and allow peroxidation to be monitored in multiwell plate readers [57].

### 5.3. Aqueous Versus Lipid-Phase Antioxidant Assays

The oxygen radical absorbance capacity (ORAC) assay is performed in an aqueous buffered system and evaluates antioxidant protection against ROO^•^–mediated oxidation, primarily reflecting antioxidant activity in water-phase biological environments such as the cytosol or plasma [58]. In its classical format, the fluorescent probe (typically fluorescein) is a freely diffusing, water-soluble molecule, and oxidation occurs in a homogeneous aqueous phase, limiting the relevance of ORAC for lipid peroxidation processes taking place in membranes or lipid-rich microenvironments [59].

To partially address this limitation, Varandas et al. introduced a liposome-adapted ORAC format in which the fluorophore was covalently anchored to a phospholipid (POPE–COUM) and incorporated into lipid bilayers [60]. This configuration spatially confines oxidative events to the membrane or interfacial region, providing a more biomimetic context than classical ORAC. However, the system still relies on ROO^•^/RO^•^ radicals generated by thermal decomposition of AAPH in the aqueous phase and does not involve intrinsic lipid autoxidation [61]. Accordingly, such approaches should be regarded as modified ORAC rather than genuine lipid autoxidation models.

Overall, ORAC and its variants offer useful reference information for antioxidant screening in aqueous or membrane-associated settings, but they may not adequately capture the kinetic features of lipid radical chain propagation that dominate the antioxidant behavior of lipophilic EO constituents. In contrast, fluorescence-enabled inhibited autoxidation (FENIX) is directly grounded in the mechanism of lipid peroxidation because the fluorescent probe is lipophilic, is co-oxidized with lipids being able to transfer the radical chain. By monitoring antioxidant interference with the radical propagation phase in controlled autoxidation systems, FENIX enables quantitative determination of radical-trapping rate constants and inhibition efficiencies [62]. Because lipid peroxidation itself serves as the reaction driver, FENIX more faithfully reflects oxidation kinetics in lipid phases and membrane environments.

Consistent with this mechanistic basis, FENIX has been increasingly applied to the study of membrane-associated antioxidant defense, particularly in ferroptosis research, where it provides quantitative insight into the suppression of phospholipid peroxidation [63,64]. For EO components whose biological effects are closely linked to membrane lipid peroxidation, FENIX therefore offers higher mechanistic relevance and predictive value than conventional aqueous antioxidant capacity assays.

### 5.4. Deprecated Methods

Many researchers commonly rely on DPPH (2,2-Diphenyl-1-picrylhydrazyl), ABTS (2,2′-Azino-bis(3-ethylbenzothiazoline-6-sulfonic acid)), and FRAP (Ferric Reducing Antioxidant Power) assays to assess antioxidant activity. However, these radicals differ chemically and mechanistically from the lipid ROO^•^ that propagate lipid peroxidation. For instance, some of these model radicals exhibit enhanced reactivity in polar solvents, while ROO^•^ radicals do not. Moreover, the common practice of measuring absorbance at a fixed time can mask kinetic differences, effectively equating the activity of “fast” antioxidants with that of “slow” ones, and potentially leading to misleading conclusions about their true efficacy in lipid systems [65].

DPPH is an excellent electron acceptor, and this differentiates it from peroxyl radicals, which are more active in HAT. Therefore, compounds that deprotonate to give their conjugated base can react very quickly with DPPH by electron transfer from the anion by the sequential proton-loss electron transfer (SPLET) mechanism. Furthermore, the use of DPPH under non-kinetic conditions with the aim of determining its consumption at fixed times may overestimate the antioxidant capacity of a compound. Overall, the DPPH assay gives the scavenging activity or the reducing capacity of a compound, but it does not provide an appropriate measurement of the antioxidant activity. However, it is important to note that, despite the limitations of the DPPH assay, it can be useful for studying structure-activity relationships among antioxidants and for titrating the content of antioxidants in crude extracts, provided that the reaction time is short to minimize errors.

## 6. Biological Methods and Mechanistic Evaluation System

A comprehensive investigation of the antioxidant activities of EOs requires the systematic mapping of the molecular networks and signaling pathways through which they exert their protective effects. Moving beyond classic in vitro assays, contemporary research adopts a multidimensional framework that combines biochemical, cellular, and in vivo analyses with cutting-edge high-throughput screening, target-specific profiling, and molecular interaction platforms to enable a comprehensive evaluation of antioxidant capacity.

### 6.1. Detection and Quantification of Oxidative Mediators and Byproducts

#### 6.1.1. Fluorescent and Chemiluminescent Probes for Reactive Species

Intracellular levels of free radicals, nitric oxide, and hydrogen peroxide are established markers of oxidative stress and are routinely measured to assess the antioxidant capacity of EOs. Recent methodological advances have led to the development of a diverse array of cell-permeable fluorescent probes, each with selectivity for distinct reactive oxygen or nitrogen species, enabling sensitive and dynamic monitoring within biological systems. Table 1 summarizes the probes used in studies on EOs and compounds.

Reactive oxygen species (ROS) in cells arise from multiple biochemical processes, among which electron leakage from the mitochondrial respiratory chain constitutes a major endogenous source under both physiological and pathological conditions. Additional contributions originate from enzymatic systems, including NADPH oxidases, xanthine oxidase, and uncoupled nitric oxide synthase, as well as from peroxisomal metabolism and inflammation-associated reactions [66]. Because individual ROS differ markedly in reactivity, lifetime, and subcellular localisation, their accurate assessment requires probes with appropriate selectivity and compartmental resolution, and no single probe can adequately represent the overall cellular redox state [67].

For general ROS measurement, DCFH-DA is frequently employed [68]. MitoSOX™ Red and MitoTracker^®^ Red CM-H_2_XRos are commonly used for mitochondrial ROS detection [69,70]. While dihydroethidium (DHE) and MitoSOX™ Red are widely applied for superoxide detection, their selectivity remains a matter of debate due to possible oxidation by other reactive species. Likewise, Amplex^®^ Red and peroxy-orange 1 (PO1) are employed for hydrogen peroxide measurement, though the assay depends on the formation of radical intermediates via peroxidase-mediated reactions [71,72]. 4-hydroxyphenyl fluorescein (HPF) and 2-[6-(4′-amino)phenoxy-3H-xanthen-3-on-9-yl]benzoic acid (APF) are utilized for hydroxyl radicals [73]. DAF-FM DA (i.e., 4-amino-5-methylamino-2′,7′-difluorofluorescein diacetate) is specific for nitric oxide [74], and Dihydrorhodamine 123 (DHR123) is applied for peroxynitrite (ONOO^−^) [75]. C11-BODIPY 581/591 is widely adopted for monitoring lipid peroxidation [76]. Genetically encoded biosensors, such as the fluorescent protein–based probe HyPer, enable real-time monitoring of hydrogen peroxide in living cells and allow dynamic, compartment-specific tracking of its fluctuations [77].

While these tools have significantly advanced the resolution and specificity of redox monitoring (Figure 7), each probe has inherent limitations regarding selectivity, photostability, or susceptibility to artifacts. Careful experimental validation is therefore required for a reliable data interpretation.
antioxidants-15-00037-t001_Table 1Table 1Fluorescent probes are used in EO studies for detecting free radicals and oxidants in biochemical assays.ProbesTarget SpeciesEOs/CompoundsAntioxidant EffectsRefDCFH-DAIntracellular total ROSLavender EOSignificantly reduced intracellular ROS levels in H_2_O_2_-treated PC12 cells[78]DHESuperoxide (O_2_^•−^) (mainly cytosolic)Citronella and Nutmeg EOsSignificantly reduced ROS levels in the ankle joints of monosodium urate-induced gouty arthritis mice[79]MitoSOX RedMitochondrial superoxide (O_2_^•−^)Lippia alba EOSignificantly decreased mitochondrial superoxide levels in J774A.1 murine macrophage[80]Amplex™ RedHydrogen peroxide (H_2_O_2_)*Citrus aurantifolia* EOSignificantly reduced H_2_O_2_ levels in dystrophic muscle cells[81]C11-BODIPY 581/591Lipid peroxidesγ-TerpineneEffectively inhibits lipid peroxidation and protects SH-SY5Y cells from RSL3-induced ferroptosis[10]HPF/HPF-DAHydroxyl radicals (^•^OH)Pomelo peel EOSignificantly attenuated ·OH and overall ROS accumulation in both in vivo and in vitro models of cerebral ischemia–reperfusion injury[82]DAF-FM DANitric oxide (NO)CarvacrolSignificantly increased NO levels in rat cavernous endothelial cells under D-(+)-galactose-induced premature senescence[83]

#### 6.1.2. Metabolic and Oxidative Byproduct Quantification

The overproduction of reactive species ultimately leads to the formation of a range of metabolic and oxidative byproducts, which serve as important biomarkers of oxidative stress. A thorough understanding of oxidative stress and antioxidant intervention relies on accurate measurement of metabolic and oxidative byproducts within biological systems. For example, the accumulation of lipid peroxidation end-products such as malondialdehyde (MDA), 4-hydroxynonenal (4-HNE), and F2-isoprostanes has been used as a direct indicator of membrane oxidative damage (see Figure 8) [84]. Quantification of these markers, along with protein carbonyls and oxidized nucleotides like 8-hydroxy-2′-deoxyguanosine (8-OHdG), has advanced significantly with the adoption of analytical platforms such as LC-MS/MS and HPLC-FLD [85]. These techniques offer substantial improvements in selectivity and sensitivity compared to classical spectrophotometric or colorimetric assays, and they help reduce the background interference that often complicates measurements in complex biological samples. As shown in Table 2, studies evaluating oxidative byproducts with a focus on essential oils and their constituent compounds are summarized. antioxidants-15-00037-sch008_Scheme 8Scheme 8Biosynthetic pathways of MDA, 4-HNE, 8-OHdG, and F_2_-isoprostanes. (**A**) MDA and 4-HNE are generated from the peroxidation of polyunsaturated fatty acids (PUFAs); (**B**) 8-OHdG is formed through oxidative modification of deoxyguanosine residues in DNA under conditions of oxidative stress; (**C**) 8-iso-PGF_2_α is produced from arachidonic acid via free radical-catalyzed lipid peroxidation.
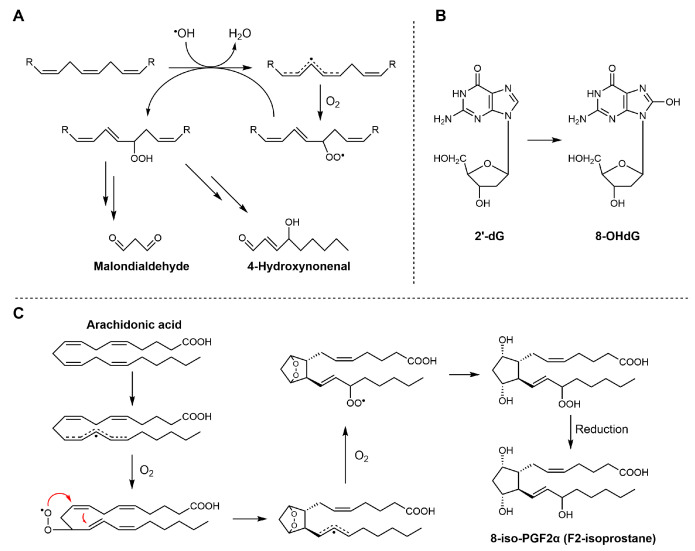


However, the selection of biomarkers and analytical methods has some limitations. For example, the quantification of MDA using the conventional thiobarbituric acid reactive substances (TBARS) assay often results in an overestimation of oxidative damage, primarily due to insufficient specificity [86]. In this assay, a range of reactive carbonyl species, some of which are not derived from lipid peroxidation, can react with thiobarbituric acid, thereby contributing to the final signal. In addition, the sensitivity of the absorbance-based TBARS assay is relatively low, with a detection limit of about 1.1 μM. The results are highly subject to variations in experimental procedures and pre-analytical sample handling. Small changes in storage temperature or sample preparation can significantly influence the outcome, making the assay poorly reproducible and unsuitable for direct comparisons between laboratories [87]. On the other hand, mass spectrometry-based methods provide superior accuracy and specificity in the quantification of oxidative markers. However, these approaches require more extensive sample preparation and access to advanced instrumentation [88].

In contrast, measuring 4-HNE and F2-isoprostanes is considered more suitable for assessing lipid peroxidation. Both molecules are generated almost exclusively through free radical-mediated oxidation of polyunsaturated fatty acids. Specifically, 4-HNE mainly arises from the peroxidation of n-6 polyunsaturated fatty acids.

F2-Isoprostanes, formed via non-enzymatic free radical oxidation of arachidonic acid, are widely regarded as the “gold standard” biomarkers of in vivo lipid peroxidation, due to their strong association with oxidative damage and various diseases. The compound 8-OHdG (see Figure 6), often produced in parallel with lipid peroxidation, is widely used as a biomarker of oxidative DNA damage. It reflects oxidative modifications of nucleic acids induced by hydroxyl radicals (^•^OH). Quantification of these biomarkers is usually performed using high-performance liquid chromatography techniques, which enable precise discrimination of structural isomers and minimize interference from unrelated compounds. However, these biomarkers are typically present at low concentrations, and their measured levels can be influenced by sample handling and storage conditions prior to analysis.
antioxidants-15-00037-t002_Table 2Table 2EO studies involving metabolic oxidative biomarkers and evaluation methods.EosModelOxidative BiomarkerMethodRef*Mentha piperita*CCl_4_-induced hepatic oxidative damage and renal failure in ratsLiver MDAThiobarbituric acid assay[89]*Lavandula stoechas*Alloxan-induced diabetic ratsLiver and kidney MDAThiobarbituric acid assay[90,91]*Rosmarinus officinalis*CCl_4_-induced acute liver damage in ratsLiver MDAThiobarbituric acid assay[92]*Origanum vulgare*Day-old chickens for 38 daysBreast and thigh muscle MDAThiobarbituric acid assay[93]*Citrus aurantifolia*Dystrophic muscle cells4-HNE-protein adductsWestern blot[81]*Lavandula angustifolia*The human glioblastoma U87MG cell line4-HNEImmunofluorescence[94]*Melaleuca alternifolia*Macrobrachium rosenbergii 4-HNEELISA[95]*Ocimum basilicum*, *Galium odoratum*, *Cymbopogon citratus*Human lymphocytes8-OHdGELISA assay[96]*Origanum vulgare*IFN-γ and histamine-induced inflammatory model in human keratinocytes (NCTC 2544)8-OHdGImmunofluorescence[97]*Thymus vulgaris*Heat stress and dietary supplementation model in laying hens8-OHdGELISA assay[98]*Origanum vulgare*Dietary supplementation model in mature Duroc boars8-OHdGELISA assay[99]*Origanum vulgare*Perinatal dietary intervention model in sows8-OHdGELISA assay[100]*Origanum majorana*Parental and epirubicin-resistant human lung cancer (H1299) cell lines8-OHdGELISA assay[101]*Lippia sidoides*Porcine pancreatic elastase-Induced Emphysema in Mice8-iso-PGF2αImmunofluorescence[102]

To move beyond single-marker analyses, recent research has embraced untargeted metabolomics and lipidomics, enabling comprehensive profiling of oxidative alterations and metabolic shifts following EO administration [103,104,105]. These high-throughput, data-rich approaches have the potential to uncover unexpected biomarkers, trace the metabolic fate of EO-derived compounds, and thereby elucidate the broader impact of oxidative stress on cellular metabolism [106]. At the same time, the vast quantity of generated data presents new challenges, including those related to data interpretation and normalization. Critical evaluation of results remains essential to draw meaningful conclusions regarding EO efficacy and underlying mechanisms.

### 6.2. Robust In Vitro and In Vivo Models for Antioxidant Efficacy

#### 6.2.1. Cellular and Organotypic Models

Cellular models remain at the heart of experimental antioxidant research, offering a controlled environment to probe the effects of EOs on oxidative stress. Most studies begin with conventional two-dimensional cell cultures, either using immortalized lines or primary cells, and expose them to oxidative insults. Typical inducers include H_2_O_2_, which is valued for its simplicity and rapid ROS elevation [107]; AAPH, a generator of peroxyl radicals, is often preferred for mimicking lipid oxidation [108]. In contrast, newer agents such as RSL3 and ML210 have gained popularity for their ability to trigger ferroptosis by targeting GPX4 activity, thereby inducing lipid peroxidation [109]. Table 3 summarizes the application of various cell-based oxidative stress models, induced by diverse agents, in evaluating the antioxidant activity of EOs.

The in vitro research on the antioxidant properties of EOs has traditionally been focused on two-dimensional (2D) monolayer cell cultures. However, as cell culture technologies evolved, the use of three-dimensional (3D) models such as spheroids and organoids has made it possible to replicate the in vivo environment for the evaluation of EO activity [110]. Unlike 2D cultures, 3D models may include some key features of real tissues, including spatial organization, dynamic cell-to-cell communication, and physiological gradients of nutrients and oxygen [111]. For instance, Azadi et al. [112] used multicellular spheroid models to demonstrate that *Zataria multiflora* EO could suppress apoptosis in spheroids formed from human breast cancer 4T1 and mouse cervical cancer TC1 cells, suggesting that 3D platforms could be extended to study redox-related EO activities in a physiologically relevant setting.

Among available 3D models, organoids are remarkable for their structural complexity and functional relevance. Derived from stem cells, organoids contain multiple cell types and exhibit an organized tissue architecture, which enables them to replicate physiological and pathological processes. For example, Bejoy et al. [113] used human kidney organoids to model acute kidney injury and found that ascorbate could reduce oxidative damage. Nevertheless, studies investigating the effects of EOs in organoid models remain limited. In summary, three-dimensional cultures reflect in vivo conditions better than conventional 2D assays, providing a valuable platform for studying the antioxidant mechanisms of EOs and facilitating future clinical applications (Figure 4).
antioxidants-15-00037-t003_Table 3Table 3Representative cellular oxidative stress inducers, mechanisms, and simulated models.MoleculesMechanism and Biological ModelEOs/CompoundsProtective MechanismsRefAAPHInduces lipid peroxidation and oxidative stress, and is widely employed as a standard probe in antioxidant screening assays.Cinnamon, Thyme, Clove, Lavender, PeppermintRadical-trapping chain-breaking activity toward peroxyl radicals[114]H_2_O_2_Elevates intracellular ROS levels, thereby inducing general oxidative stress and serving as a model for neurodegeneration and ischemia–reperfusion injury.*Alpinia zerumbet*Direct scavenging of intracellular ROS, preservation of cellular GSH levels, and protection against oxidative DNA damage[115]tert-Butyl hydroperoxide (t-BHP)Generates ROS and initiates lipid and protein oxidation, providing a reliable model for evaluating oxidative damage and cellular antioxidant defenses.Thymol, CarvacrolAttenuation of lipid peroxidation and cytotoxicity in normal fibroblasts[116,117]RSL3, ML210, ErastinInhibit GPX4 or system Xc^−^, inducing ferroptosis and enabling the study of ferroptotic cell death in contexts such as neurodegeneration, cancer, and kidney injury.Pomelo Peel, *Amomum kravanh*, β-CaryophylleneActivation of the Nrf2 signaling axis, upregulation of GPX4 and SLC7A11, suppression of phospholipid peroxidation and inhibition of ferroptosis[118,119,120,121]

#### 6.2.2. Animal Models of Oxidative Stress and Disease

Animal models are indispensable for elucidating the role that oxidative stress has in tissue injury and also for evaluating the therapeutic efficacy of antioxidants in vivo. These models allow for an assessment of pharmacokinetic and pharmacodynamic properties, including absorption, distribution, metabolism, excretion, and finally physiological and pathological responses in the organism [122,123]. Pathological conditions in oxidative stress resemble those observed in human diseases. Animal models facilitate mechanistic investigations and the identification of relevant molecular targets [124,125]. Furthermore, they provide critical platforms for screening and validating antioxidants in terms of their bioavailability and in vivo efficacy [126].

Animal models are generally utilized following the addition of agents that directly or indirectly promote the accumulation of ROS or trigger oxidative metabolic disturbances [124]. As an example, liver damage associated with oxidative stress can be reproduced using carbon tetrachloride (CCl_4_) or *tert*-butyl hydroperoxide (*t*-BHP) [127,128], while doxorubicin (adriamycin) is commonly used to study heart injury associated with excessive ROS production [129]. In the context of the nervous system, exposure to paraquat or 1-methyl-4-phenyl-1,2,3,6-tetrahydropyridine (MPTP) is used to induce oxidative stress-related neurotoxicity [130]. Acute lung injury can be induced by lipopolysaccharide (LPS) injections or prolonged cigarette smoke exposure, which initiate inflammatory oxidative cascades [131,132]. Streptozotocin (STZ) is a classic tool for generating diabetic models, in which oxidative damage is a major contributor to disease progression [133]. High-fat diets are typically used to cause sustained oxidative stress in studies of metabolic syndrome, obesity, or atherosclerosis [134]. Finally, additional protocols include H_2_O_2_, AAPH, radiation, or ischemia–reperfusion injury to selectively target certain organs or systems [135,136,137].

In recent years, a growing body of in vivo research has utilized various chemically induced oxidative stress models to evaluate the antioxidant properties of EOs and their bioactive components (Table 4). By employing diverse animal models, each characterized by specific chemical agents, the studies have shown the protective effects of EOs in a wide range of pathological conditions and provide experimental support for the therapeutic potential of EOs in the management or prevention of oxidative stress–related diseases.

The selection of animal models and oxidative stress induction methods profoundly influences the interpretation and translational relevance of in vivo studies on antioxidant activity. Chemically induced systems, although widely used, reproduce only limited aspects of oxidative stress rather than its full physiological complexity. For example, agents such as paraquat or streptozotocin trigger acute oxidative injury through redox cycling or mitochondrial disruption but may bypass the regulatory networks that underlie chronic, multifactorial diseases [138]. Moreover, substantial variability in strain susceptibility, dosing regimens, and assessment parameters can result in inconsistent phenotypes and hinder cross-study comparisons. These models may also be confounded by direct toxic effects unrelated to disease progression, and their inability to reproduce complex pathological hallmarks such as progressive fibrosis or inflammation further limits their translational value [139]. Consequently, the use and interpretation of animal models require a mechanistically informed and cautious approach to derive biologically meaningful insights into the antioxidant mechanisms of EOs.

Recently, several technological improvements have made it possible to observe oxidative events in animal models with great precision. For example, real-time imaging approaches such as bioluminescence [140] and positron emission tomography (PET), enable researchers to trace oxidative stress-related probes or drugs within the body, giving a non-invasive window into molecular processes [141]. The use of animals engineered to express biosensors that respond to redox changes has further expanded experimental techniques.
antioxidants-15-00037-t004_Table 4Table 4Representative cellular oxidative stress inducers, mechanisms, and simulated models in vivo.InducerInduced ModelsOxidative MechanismEOs/CompoundsEffects In VivoRefCarbon tetrachloride (CCl_4_)CCl_4_-induced hepato/renal toxicity in miceRadical metabolite causes hepatic lipid peroxidation and injury*Salvia officinalis* EOReduced liver/kidney damage, oxidative stress, DNA breaks; improved antioxidant defenses and tissue structure[142]CisplatinCisplatin-induced nephrotoxicity in Balb/c miceMitochondrial dysfunction, renal oxidative stress*Pituranthos chloranthus* EOMitigated cisplatin-induced nephrotoxicity, reduced DNA damage, oxidative stress, and inflammation[143]Doxorubicin (Adriamycin)Doxorubicin-induced nephrotoxicity in male Wistar ratsROS production, cardiomyocyte/liver/kidney injury*Satureja khuzistanica* EOAttenuated DOX-induced nephrotoxicity and apoptosis via mitochondrial/extrinsic pathway; limited effect on oxidative stress[144]D-galactoseD-galactose -induced cognitive deficits in miceChronic systemic oxidative stress, mimics agingLavender EO and LinaloolImproved cognition, restored Nrf2/HO-1, SOD, GPX, and synaptic plasticity proteins[145]Lipopolysaccharide (LPS)LPS-induced acute lung injury in ratsImmune activation, inflammation-induced ROS*Chimonanthus nitens* EOReduced inflammation, improved antioxidant enzymes, attenuated lipid oxidation, modulated SCFAs[146]AcrylamideAcrylamide-induced liver toxicity in ratsNeurotoxicity via ROS generation*Thymus satureioides* EOSuppressed liver enzymes, oxidative stress, NLRP3 inflammasome/NF-κB axis, and collagen deposition[147]High-fat diet (HFD)HFD-induced nonalcoholic fatty liver disease in micePromotes lipid peroxidation, metabolic stressGinger EOAmeliorated hepatic injury, improved lipid metabolism, suppressed oxidative stress and inflammation[148]Monosodium urate (MSU)MSU-induced gouty arthritis in ratsCrystal-induced inflammation and ROSCitronella and Nutmeg EOsReduced joint swelling, neutrophil infiltration, oxidative stress, and NLRP3 inflammasome activity[79]Isoproterenol (ISO)ISO-induced myocardial infarction in ratsCardiac oxidative stress, induces heart injury*Commiphora molmol* EOAmeliorated cardiac injury, improved Nrf2/HO-1 pathway, suppressed oxidative stress, inflammation, and apoptosis[149]Streptozotocin (STZ)STZ-nicotinamide-induced type 2 diabetes in ratsBeta-cell destruction, induces diabetes and oxidative stress*Mentha piperita* EOReduced hyperglycemia, improved insulin/C-peptide, enhanced antioxidant status, protected liver and pancreas tissues[150]ParaquatParaquat-induced pulmonary toxicity in ratsRedox cycling, massive ROS generationMyrtenolRestored SOD, CAT, thiol, reduced TNF-α, IL-6, MDA, improved lung histopathology and antioxidant status[151]EthanolEthanol-induced gastric injury in ratsInduces liver oxidative stress and lipid peroxidation*Rosmarinus officinalis* EOProtected against gastric lesions via antioxidant action; increased SOD and GSH-Px, reduced lipid peroxides[152]Ultraviolet (UV) radiationUVB-induced oxidative stress in albino ratsDirectly increases ROS, DNA and skin damage*Calendula officinalis* EOSignificantly decreased MDA, increased catalase, GSH, SOD, ascorbic acid, and total protein in skin tissue[153]Mercury chloride (HgCl_2_;)HgCl_2_;-induced oxidative damage in male rats (testis, spleen, kidney)Induces oxidative stress, inflammation, and reproductive/renal toxicity*Origanum* EODecreased TBARS, increased GSH, SOD, and CAT; restored trace elements (Zn, Cu, Mg, Fe), testosterone, and improved histological alterations in testis and spleen[154]

### 6.3. Molecular Pathways and Redox Regulation Networks

#### 6.3.1. Core Redox-Signaling Pathways and Experimental Approaches

The Nrf2/Keap1 pathway acts as a central regulator of the antioxidant defense by regulating the expression of numerous cytoprotective genes [68]. The NF-κB pathway complements this role by modulating inflammatory and immune responses, thereby linking redox regulation with inflammation and cellular protection [155]. Mitogen-activated protein kinase (MAPK) signaling, through cascades including ERK, JNK, and p38 MAPK, further orchestrates cellular decisions involving survival, proliferation, differentiation, and apoptosis in response to oxidative cues [156]. Additionally, the recently characterized ferroptosis pathway represents a crucial iron-dependent mechanism of regulated cell death associated with lipid peroxidation [157]. This has paved the way for new studies aimed at understanding oxidative damage and cellular homeostasis.

To dissect these intricate signaling networks, a broad array of experimental approaches has been employed. Immunoblotting and immunofluorescence techniques facilitate the visualization and quantification of protein expression levels, phosphorylation status, and nuclear translocation events. Quantitative reverse transcription PCR (qRT-PCR) allows precise quantification of transcriptional changes in genes involved in redox signaling. Luciferase reporter assays provide a functional assessment of transcription factor activity, which enables direct evaluation of promoter activation events. Furthermore, cutting-edge phosphoproteomic profiling offers extensive mapping of protein phosphorylation events, revealing critical post-translational modifications that are pivotal in signal transduction and redox regulation. These sophisticated methodologies enable the comprehensive analysis and interpretation of the dynamic regulatory events underpinning EO-induced antioxidant responses.

#### 6.3.2. Activation of the Nrf2/Keap1 Signaling Pathway

The Nrf2/Keap1 pathway is fundamental for protecting cells from oxidative and electrophilic stress. Nrf2 acts as a transcription factor that induces a wide spectrum of antioxidant and detoxifying genes. In resting conditions, Nrf2 is retained in the cytoplasm by Keap1, a sensor protein rich in reactive cysteine residues [158]. Upon exposure to oxidative or electrophilic stimuli, critical cysteine residues on Keap1 are chemically modified, which weakens its ability to repress Nrf2. As a consequence, Nrf2 becomes stabilized, translocates into the nucleus and binds to antioxidant response elements (AREs), thereby initiating the transcription of downstream cytoprotective genes, including HO-1, SOD, CAT, GPX and NQO1 [159].

The proteins encoded by these genes collectively contribute to cellular redox control by limiting the accumulation of ROS/reactive nitrogen species (RNS). Through the coordinated actions of SOD, CAT, GPX and HO-1, highly reactive species such as superoxide (O_2_^•−^), hydrogen peroxide (H_2_O_2_), peroxynitrite (ONOO^−^) and hydroxyl radicals (^•^OH) are efficiently detoxified. Beyond direct ROS scavenging, Nrf2 signaling also interfaces with iron redox metabolism and the glutathione system (GSH/GSSG/NADPH), forming an integrated network that supports redox homeostasis and enhances cellular resistance to oxidative injury (Figure 5).

The pathway can be triggered either directly, through covalent modification of reactive cysteine residues on Keap1 by electrophilic or oxidative compounds, or indirectly, through upstream signaling cascades such as MAPK or PI3K/Akt that facilitate Nrf2 release from Keap1 [161]. Structurally, Keap1 is highly sensitive to electrophilic molecules [162]. As summarized in Table 5, cinnamaldehyde is the most extensively studied EO constituent containing an α,β-unsaturated carbonyl group. Other EO compounds, including citral, carvone, and pulegone, share this structural feature and exhibit antioxidant activity via Nrf2 pathway modulation. This moiety is present in many phytochemicals capable of undergoing Michael-type addition with thiol groups on Keap1 cysteines [163]. In addition, compounds such as turmerones from turmeric EO, as well as β-ionone, β-damascenone, jasmone, and irones, possess similar carbonyl functionalities. Although these compounds are structurally predisposed to interact with Keap1 (Figure 6), direct evidence for their Nrf2-related activity is still lacking, making them promising candidates for future investigation.

To date, there is no direct evidence demonstrating that EO-derived molecules can bind to Nrf2 or Keap1. Experimental techniques such as surface plasmon resonance, isothermal titration calorimetry, X-ray crystallography, or pull-down assays have rarely been applied. Most studies rely on indirect functional assays, including the monitoring of Nrf2 nuclear translocation or changes in downstream antioxidant enzymes, sometimes combined with inhibitors or siRNA. While molecular docking has been employed to predict potential interactions, these computational findings have seldom been validated experimentally. As a result, definitive proof of direct binding between EO constituents and these proteins remains elusive.

Many EO constituents can also activate the Nrf2 pathway through electrophilic interactions, as summarized in Table 5. Phenolic compounds such as carvacrol, thymol, and eugenol exhibit potent antioxidant properties but may also undergo oxidation to yield electrophilic quinone-like intermediates. These reactive metabolites are capable of covalently modifying Keap1 cysteine residues, thereby disrupting the Keap1–Nrf2 complex and promoting Nrf2 nuclear translocation. This direct mechanism explains the consistent upregulation of Nrf2-regulated antioxidant genes observed following exposure to these phenolic EO constituents. Non-phenolic constituents, including monoterpenes such as β-caryophyllene, geraniol, linalool, limonene, 1,8-cineole, α-pinene, and p-cymene, also contribute to cellular antioxidant regulation. Although these molecules lack electrophilic α,β-unsaturated carbonyl groups, they may still modulate Nrf2 activity by forming electrophilic metabolites during phase I biotransformation or ROS-mediated oxidation [164]. The mechanisms involved are not yet fully understood and warrant further study. Overall, both phenolic and non-phenolic EO components can shift cellular redox conditions to favor Nrf2 activation. However, current conclusions remain based mainly on chemical reactivity and cell-based assays, whereas strong direct evidence from EO systems is still scarce.
antioxidants-15-00037-t005_Table 5Table 5Effects of EOs and their constituents on the Nrf2 pathway in different oxidative stress models and corresponding detection methods.Compounds/EOsInduced ModelRegulation of the Nrf2 PathwayDetection MethodsRefCinnamaldehydeH_2_O_2_-induced oxidative stress in HepG2 cellsPromotes nuclear translocation of Nrf2 and upregulates downstream antioxidant enzymes.Western blot, immunofluorescence, qPCR, siRNA[165]CinnamaldehydeHigh-glucose mouse aorta and endothelial cellsEnhances nuclear translocation and expression of Nrf2, and increases HO-1 and NQO1 expression.Western blot, immunofluorescence, siRNA[166]Cinnamaldehydedb/db diabetic mouse aorta and kidney tissuesUpregulates Nrf2, HO-1, GPX-1, and NQO-1, and ameliorates oxidative damage.Western blot, qPCR, siRNA[167]CinnamaldehydeH_2_O_2_-treated bone marrow mesenchymal stem cells and ovariectomized miceIncreases Nrf2, HO-1, and NQO-1 expression, and promotes Nrf2 nuclear translocation.Western blot, immunofluorescence, immunohistochemistry (IHC), siRNA[168]CinnamaldehydeH_2_O_2_-treated human dermal papilla cellsPromotes Nrf2 nuclear translocation and increases HO-1 expression.Western blot, immunofluorescence, qPCR, siRNA[169]CinnamaldehydeBaP-induced oxidative stress in HaCaT cellsPromotes Nrf2 nuclear translocation and increases HO-1 expression.Western blot, immunofluorescence, qPCR, siRNA[170]CinnamaldehydeHuman umbilical vein endothelial cellsUpregulates Nrf2 downstream antioxidant enzymes.Western blot, immunoprecipitation, nuclear fractionation, siRNA[171]CinnamaldehydeNormal human epidermal keratinocytesUpregulates GPX2 and NQO1 in an Nrf2-dependent manner.Western blot, siRNA[172]CinnamaldehydeTGF-β1/IL-13-treated normal human dermal fibroblastsPromotes nuclear translocation of Nrf2 and increases HMOX1 and NQO1 expression, exerting antifibrotic effects.Western blot, immunofluorescence, qPCR, siRNA[173]CinnamaldehydeHCT116 colorectal cancer cells and mouse colon tissuesUpregulates Nrf2, HO-1, and NQO1, and downregulates Keap1.Western blot, qPCR, IHC[174]CinnamaldehydeH_2_O_2_-treated V79-4 lung fibroblastsIncreases Nrf2, phospho-Nrf2, and HO-1 expression, and decreases Keap1.Western blot, ELISA, siRNA[175]CinnamaldehydeHigh-fat and high-glucose diet-induced metabolic syndrome in Wistar ratsIncreases vascular Nrf2 activity and upregulates HO-1 and related enzymes.Transcription factor DNA-binding assay, qPCR[176]CinnamaldehydeH9c2 cardiomyocytes and doxorubicin-injured rat myocardiumPromotes Nrf2 nuclear translocation and HO-1 expression, and inhibits ferroptosis.Western blot, qPCR, IHC, siRNA[177]CinnamaldehydeZymosan-stimulated KERTr human keratinocytesUpregulates Nrf2, HO-1, and NQO1 at low doses, and exerts anti-inflammatory effects in a Nrf2-dependent manner.Western blot, qPCR, shRNA, ELISA[178]CinnamaldehydeH_2_O_2_- and TNF-α-stimulated human umbilical vein endothelial cellsPromotes Nrf2 nuclear translocation, upregulates HO-1, and reduces inflammation.Western blot, qPCR, siRNA, immunoprecipitation[179]CinnamaldehydeLPS-induced neuroinflammation mouse modelPromotes Nrf2 nuclear translocation, increases SOD and GST, and reduces MDA levels.ELISA, enzyme activity assay, Western blot[180]CinnamaldehydeOxygen-glucose deprivation/reperfusion-injured H9c2 cardiomyocytesUpregulates Nrf2, HO-1, and PPAR-γ expression.Western blot, qPCR, siRNA, enzyme activity assays[181]CinnamaldehydeIn vitro thioredoxin reductase assay using HCT116 colorectal cancer cellsActivates the Nrf2/ARE pathway, upregulates TrxR, and inhibits TrxR in a dose-dependent manner.Luciferase reporter assay, Western blot, enzymology[182]CinnamaldehydeH_2_O_2_- and arsenic-treated HCT116, HT29, and FHC colon cellsIncreases Nrf2, HO-1, and γ-GCS expression, and exerts Nrf2-dependent antioxidant protection.Luciferase reporter assay, Western blot, qPCR, siRNA[183]CitralAdriamycin-induced FSGS mouse model and RAW264.7 macrophagesAlleviates renal injury by activating Nrf2, upregulating HO-1/NQO1, and inhibiting oxidative stress and apoptosis.Western blot, ELISA, IHC[184]CitralLPS-induced endometritis mouse model and Nrf2 knockout miceActivates the Nrf2/HO-1 pathway, suppresses LPS-induced ferroptosis and inflammation, and exhibits Nrf2-dependent protective effects.Western blot, siRNA[185]CitralLPS-accelerated lupus nephritis mouse model and LPS-primed macrophagesEnhances Nrf2 activation, upregulates HO-1 and GPX, and attenuates oxidative stress and NLRP3 inflammasome signaling.Western blot, IHC, Immunofluorescence, ELISA, TUNEL[186]Citral, Lemongrass EOPC12D neuronal cell in vitro modelTargets Keap1 and promotes Nrf2-dependent HO-1 expression, as shown by molecular docking and cellular experiments; animal studies confirm antioxidative effects in brain regions.Molecular docking, qPCR, Western blot[187]CarvoneLPS-induced acute lung injury in ratsSignificantly upregulates Nrf2 and HO-1 expression in lung tissue, and attenuates oxidative stress and inflammation.Western blot, histopathology, ELISA, enzyme activity assay[188]CarvoneCCl4-induced liver fibrosis in ratsUpregulates the Nrf2 pathway, improves antioxidant status (increases GSH, SOD), reduces oxidative damage and fibrosis, and is associated with reduced TGF-β1/SMAD3 signaling.Enzyme activity assay, IHC, qPCR, liver histology[189]PulegoneLPS-stimulated RAW264.7 macrophagesPromotes Nrf2/HO-1 expression, downregulates iNOS, COX-2, NF-κB, and MAPKs, and suppresses inflammation and ROS.Western blot[190]PulegoneL-arginine-induced acute pancreatitis mouse modelInhibits the p38 MAPK/NF-κB pathway, alleviates oxidative stress and inflammatory responses, and upregulates Nrf2 and antioxidant defense enzymes.Western blot[191]EugenolH_2_O_2_-induced injury models in HEK-293 and NIH-3T3 cellsDose-dependently activates Nrf2 expression and nuclear translocation, upregulates Nrf2 target genes, and exerts antioxidant effects.Western blot, qPCR, transcriptional activity assays[192]CarvacrolRotenone-induced Parkinson’s disease mouse modelUpregulates Nrf2/HO-1, reduces inflammation, oxidative stress, and NLRP3 activation, and improves motor and neural injury.Western blot, IHC, enzyme activity assay[193]CarvacrolSTZ-induced diabetes in ratsUpregulates Nrf2/HO-1, enhances antioxidant enzyme activity, and inhibits NF-κB-mediated testicular apoptosis and inflammation.Western blot, enzyme activity assay[194]Thymol, p-CymeneImmobilization stress in ratsIncreases Nrf2 and HO-1 expression, suppresses TNF-α/NF-κB and oxidative stress, and reduces liver inflammation.qPCR, ELISA, histology[195]α-PineneEthanol-induced gastric injury in ratsUpregulates Nrf2 and HO-1 mRNA, increases gastric pH, and reduces lesions and oxidative damage.qPCR, histology, enzyme activity assay[196]β-CaryophylleneHepatic ischemia–reperfusion injury in ratsUpregulates Keap1/Nrf2/HO-1/NQO1, decreases inflammation via TLR4/NF-κB/NLRP3, and reduces oxidative stress.Western blot, qPCR, IHC, ELISA, enzyme activity assay, in silico docking[197]β-CaryophylleneMCAO/reperfusion (cerebral ischemia) rats; OGD/R PC12 cellsEnhances Nrf2 nuclear translocation and HO-1, suppresses ferroptosis, and effect is blocked by Nrf2 inhibitor ML385.Western blot, inhibitor rescue, neurobehavioral tests, infarct size[120]β-CaryophylleneGlutamate-induced C6 glioma cell toxicityInduces Nrf2 nuclear translocation, improves GSH and GPX, inhibits ROS, and restores mitochondrial function in a CB2R-dependent manner.Immunofluorescence, GSH, GPX, ROS, MTT, JC-1, Western blot[198]Carvacryl acetateCerebral ischemia–reperfusion injury model in rats and H_2_O_2_-induced oxidative stress model in PC12 cellsPromotes Nrf2 expression and nuclear translocation, provides antioxidative neuroprotection, and loses protective effects with Nrf2 knockdown.Western blot, IHC, shRNA[199]GeraniolHepatic ischemia–reperfusion injury model in ratsMarkedly activates Nrf2/HO-1, upregulates antioxidant enzymes, and attenuates oxidative stress and apoptosis.Western blot, enzyme activity assay[200]GeraniolDoxorubicin-induced cardiac toxicity in ratsDose-dependently upregulates Nrf2/HO-1, exhibits antioxidant, anti-inflammatory, and anti-apoptotic activities, and provides cardioprotection.Western blot, qPCR, tissue biochemical analysis[201]GeraniolHigh-fat diet-induced atherosclerosis model in hamstersSignificantly upregulates Nrf2 and antioxidant enzymes, inhibits lipid peroxidation, and improves endothelial function.Western blot, biochemical assays[202]GeraniolRenal ischemia–reperfusion injury model in ratsActivates Nrf2/HO-1/NQO1, inhibits TLR2/4-NF-κB inflammatory signaling, and protects the kidney.Western blot, qPCR, molecular docking analysis[203]Perillyl alcoholLPS-induced RAW264.7 cells, CFA-induced arthritis in ratsAmeliorates oxidative stress and inflammation via regulation of TLR4/NF-κB and Keap1/Nrf2 pathways, increases Nrf2 and SOD2, and reduces NF-κB and iNOS.Western blot, qPCR, histology[204]Perillyl alcoholOGD/R-induced PC12 cells, Rice-Vannucci hypoxic–ischemic neonatal ratsActivates Nrf2, inactivates Keap1, and reduces oxidative stress, inflammation, and apoptosis; effect is reversed by the Nrf2 inhibitor ML385.Western blot, Nrf2 nuclear localization, ML385 inhibitor, in vivo/in vitro[205]NerolidolDoxorubicin-induced chronic cardiotoxicity in ratsModulates PI3K/Akt and Nrf2/Keap1/HO-1 pathways, and inhibits oxidative and inflammatory damage.Western blot, qPCR, enzyme activity assay[206]NerolDexamethasone-induced aging in human dermal fibroblastsActivates the Nrf2 pathway, restores collagen and hyaluronic acid, and protects against glucocorticoid-induced aging; effect is abolished by Nrf2 inhibitor.Inhibitor validation, functional assays[207]PerillaldehydeX-ray-induced intestinal injury in C57BL/6J mice, intestinal organoids, HIEC-6 cellsUpregulates Nrf2, activates antioxidant pathways, and inhibits ferroptosis; effect is abolished by the Nrf2 inhibitor ML385.Western blot, functional rescue, in vivo/in vitro models[208]LimoneneUVB-irradiated HaCaT keratinocytesEnhances nuclear Nrf2 translocation, and increases HO-1, NQO1, and γ-GCLC expression.Western blot, siRNA[209]Limonene, *Coreopsis tinctoria* EOD-galactose-induced aging and cognitive impairment model in miceEO upregulates Nrf2, suppresses NF-κB, alleviates cognitive impairment and oxidative stress; shows superior efficacy compared to limonene.Western blot, qPCR[210]LinaloolIL-1β-induced chondrocytes, DMM mouse OA modelActivates Nrf2/HO-1, reduces inflammation, protects the extracellular matrix, and blocks the NF-κB pathway.Western blot, qPCR, in vivo mouse model[211]LinaloolLiver ischemia–reperfusion in ratsActivates the Keap1/Nrf2/HO-1/NQO1 axis, and suppresses the TLR4/RAGE/NF-κB pathway.Western blot, qPCR, enzyme activity assay, docking[212]Linalool, Lavender EOD-galactose & AlCl3-induced cognitive deficit in miceUpregulates Nrf2 and HO-1 expression, and improves antioxidant and synaptic markers.Western blot, qPCR, behavior tests[145]*Achillea millefolium* EOEthanol-induced gastric ulcer in ratsUpregulates Nrf2 and HO-1 expression, exerts antioxidant, anti-inflammatory, and anti-apoptotic effects, and alleviates gastric injury.Western blot, IHC[213]1,8-Cineole, *Amomum kravanh* EOAdenine and 5/6 nephrectomy-induced chronic kidney disease in ratsUpregulates Nrf2/HO-1, inhibits ferroptosis and fibrosis, and protects the kidney.RNA-seq, Western blot[121]1,8-Cineole, *Artemisia vulgaris* EOAcetaminophen-induced liver injury in miceSuppresses Keap1, promotes Nrf2 nuclear translocation and target gene (UGT/SULT) expression.Western blot, qPCR, molecular interaction assays, translocation analysis[214]BorneolAβ-induced neurotoxicity in human neuroblastoma SH-SY5Y cellsIncreases Nrf2 and Bcl-2 expression, and attenuates oxidative stress and apoptosis.Western blot[215]*Melaleuca alternifolia* EORAW264.7 murine macrophagesUpregulates HO-1 via Nrf2-ARE activation.Western blot, qPCR, reporter assay[216]Myrrh EOIsoproterenol-induced myocardial infarction in ratsIncreases Nrf2 and HO-1, and reduces apoptosis and inflammation.Western blot, enzyme activity assay, IHC[149]*Nepeta cataria* EOAcetaminophen-induced liver injury in miceInduces Nrf2 activation, upregulates phase II enzymes (UGTs, SULTs), and reduces CYP2E1.qPCR, enzyme activity assay[217]Oregano EOH_2_O_2_-induced oxidative stress in IPEC-J2 cellsIncreases Nrf2 activation, induces SOD1, GCLC, and GSH expression, and protects against ROS; effect is reversed by Nrf2 siRNA.Western blot, qPCR, siRNA, luciferase reporter[218]Oregano EOPostoperative adhesion mouse models, in vitro barrier assaysUpregulates Nrf2 phosphorylation and downregulates NF-κB, reducing inflammation and fibrosis, as shown by oregano EO-loaded nanofiber barriers.Phosphorylation assay (Western blot), protein markers[219]*Pinus morrisonicola* EOUVB-irradiated HaCaT keratinocytesActivates Nrf2, increases HO-1 and NQO-1, and reduces ROS and cell death; protection is lost with Nrf2 knockdown.Western blot, siRNA[220]*Rosmarinus officinalis* EOH_2_O_2_-induced oxidative stress in A549 cellsActivates the Nrf2 pathway, enhances NQO-1 and HO-1 expression, and exhibits strong radical scavenging activity; supported by molecular docking with Keap1.Western blot, qPCR, molecular docking[221]*Salvia lavandulifolia* EOH_2_O_2_-induced oxidative stress in PC12 cellsActivates Nrf2, increases antioxidant enzyme activity, reduces ROS and MDA, and protects against neuronal injury.Western blot, enzyme activity assay, qPCR[222]*Schisandra chinensis* EODepression mouse model, H_2_O_2_-induced PC12 cellsIncreases nuclear translocation of Nrf2 and HO-1, reverses SOD, GSH, and CAT decline, decreases MDA, and protects neurons and exerts antidepressant effects.Immunoblot, immunofluorescence, enzyme activity assay[223]*Stahlianthus involucratus* EOApoE−/− mice with high-fat diet (atherosclerosis model); ox-LDL-induced HUVECsActivates the Nrf2 pathway, increases Nrf2, HO-1, and NQO1, decreases Keap1, reduces oxidative stress, restores mitochondrial quality, and loses effects with Nrf2 silencing.Western blot, IHC, in vitro siRNA[224]Tea tree EOMacrobrachium rosenbergii fed with different doses of tea tree EOUpregulates Keap1-Nrf2 signaling, increases antioxidant enzyme and autophagy gene expression at 100 mg/kg; strongly activates Nrf2 and inhibits autophagy at 1000 mg/kg.qPCR, Western blot, functional rescue, Pearson correlation analysis[225]*Tagetes erecta* EOMNNG-induced gastric cancer in ratsUpregulates Nrf2 and HO-1, reduces NF-κB p65 and IκBα degradation, and alleviates oxidative stress, inflammation, and apoptosis.qPCR, histology, ELISA, enzyme activity assay, IHC[226]*Thymus vulgaris* EOMDA-MB-231 triple-negative breast cancer cellsInduces Nrf2 mRNA expression and HO-1, increases ROS and MDA, reduces catalase and PON2, and triggers apoptosis.qPCR, Western blot, cell viability assays[227]*Thymus quinquecostatus* EOZebrafish oxidative stress model, in vitro antioxidant assaysActivates the Nrf2/Keap1 pathway, increases SOD1, CAT, and Hmox1, and reduces ROS and lipid peroxidation.qPCR, DPPH/ABTS/FRAP/TBARS assays, zebrafish imaging[228]Chamomile EOLPS and IFN-γ-stimulated human and mouse macrophages, and peripheral blood mononuclear cellsUpregulates Nrf2, GCL, and HO-1, eliminates ROS, and exhibits anti-inflammatory and antioxidant effects by suppressing NF-κB.Western blot, qPCR, enzyme activity assay[229]*Chimonanthus nitens* EODSS-induced colitis in miceActivates Nrf2/HO-1, inhibits MAPK/NF-κB inflammatory pathways, and enhances tight junction protein expression.Western blot, qPCR[230]Coriander EODexamethasone-induced acute liver injury in ratsDose-dependently activates Nrf2/HO-1, attenuates oxidative stress and apoptosis, and improves liver injury.Western blot, IHC[231]*Dalbergia odorifera* EOIsoproterenol-induced myocardial ischemia in ratsSignificantly upregulates Nrf2 and HO-1, reduces Caspase 3/9 levels, and provides cardioprotection.IHC[232]*Eugenia uniflora* EOIn vivo fumigation of Drosophila melanogasterUpregulates Nrf2 targets and induces both oxidative stress and antioxidant responses, exhibiting toxicity.Western blot[233]Red ginseng EOH_2_O_2_-treated HepG2 cells, CCl4-induced miceUpregulates antioxidant enzymes (SOD, CAT, GPX) and inhibits MAPK phosphorylation.Western blot, IHC, enzyme activity assay[234]Lavender, lemongrass, rosemary, chamomile EOsUVB-irradiated human dermal papilla cellsIncreases Nrf2 activation and upregulates phase II enzymes (HO-1, NQO1, GST-pi).Western blot, qPCR[235]

#### 6.3.3. Multifaceted Regulation of Redox Signaling Pathways

In addition to the Nrf2/Keap1 pathway, EOs influence a range of redox-sensitive signaling networks that are fundamental to oxidative stress regulation and cellular homeostasis. The NF-κB pathway, which plays a central role in inflammation and immune function, governs the production of pro-inflammatory cytokines and oxidative enzymes. MAPK signaling, including ERK, JNK, and p38, orchestrates cell survival, apoptosis, and adaptation to stress. EO components have been shown to modulate these kinases, thereby shaping cellular responses to oxidative and inflammatory challenges. For instance, *Zingiber striolatum* EO inhibited LPS-induced activation of NF-κB and MAPK signaling in macrophages, leading to decreased ROS production both in vitro and in vivo [236]. Similarly, α-bisabolol from chamomile EO attenuated NF-κB and MAPK activation, suppressed inflammatory mediator release, and limited oxidative stress, while also enhancing Nrf2-mediated antioxidant defenses [237]. These findings highlight the interconnected nature of redox and inflammatory pathways and the multifaceted actions of EO constituents.

Ferroptosis is a form of programmed, iron-dependent cell death characterized by the accumulation of lipid peroxides, leading to loss of membrane integrity and cell viability. It has also been identified as a key redox-regulated process. It is now recognized as a central redox-regulated process driven by the failure of lipid peroxide detoxification, particularly through the GPX4–GSH axis, and by dysregulated iron homeostasis. In this context, EOs may influence ferroptosis primarily by modulating GPX4 activity, altering cellular iron metabolism, and attenuating lipid ROS accumulation, thereby interfering with the core biochemical events of ferroptotic execution [238]. For example, pomelo peel EO protected against cerebral ischemia–reperfusion injury by activating Nrf2 signaling, increasing SLC7A11 and GPX4 expression and decreasing intracellular iron and lipid peroxidation [120]. *Artemisia argyi* EO alleviated bisphenol A–induced liver damage by suppressing ferroptosis and upregulating antioxidant pathways [239]. Tea tree EO improved hepatopancreatic antioxidant status by increasing GPX4 levels in aquatic models [225]. Despite the currently limited direct evidence, the high lipophilicity and low molecular weight of EO constituents favor their partition into lipid bilayers, where they may directly interfere with lipid peroxidation dynamics and redox balance [240]. This physicochemical behavior suggests a plausible route through which volatile compounds may modulate ferroptosis at the membrane level, a hypothesis that warrants systematic investigation.

Beyond these well-established mechanisms, EOs also modulate additional regulatory axes, notably the PI3K/Akt/mTOR pathway. Recent studies have demonstrated that EOs from citronella, nutmeg, and *Matricaria chamomilla* can suppress PI3K/Akt/mTOR activation, leading to reduced inflammation and oxidative stress in models of gouty arthritis and psoriatic skin inflammation. Inhibition of this pathway by EOs was associated with decreased pro-inflammatory cytokine production and diminished ROS generation, further supporting their role in maintaining cellular redox balance [79,241]. The involvement of related networks, such as sirtuins and hypoxia-inducible factors, adds further complexity to the antioxidant potential of these natural compounds [242].

Despite substantial progress, much of the existing research remains descriptive and is largely limited to observations of pathway activation or inhibition. There is a need for deeper mechanistic studies that directly confirm molecular targets, clarify structure–activity relationships, and elucidate the biochemical basis of the antioxidant and anti-inflammatory effects of EOs. Addressing these gaps will be crucial for realizing the full therapeutic potential of EOs in the context of redox biology and inflammation-related disorders.

#### 6.3.4. Integrated Functional Evaluation of Antioxidant Enzyme Activities

Beyond classical pathway analysis, it is essential to functionally characterize the major antioxidant enzymes to understand how EOs shape cellular redox balance. Central to this defense system are enzymes such as superoxide dismutase (SOD), catalase (CAT), GPX, heme oxygenase-1 (HO-1), and NAD(P)H quinone oxidoreductase 1 (NQO1). These enzymes form a coordinated network that neutralizes ROS and preserves redox homeostasis [243].

Each enzyme fulfills a unique yet interconnected function. SOD rapidly converts superoxide anions to hydrogen peroxide [244], which is subsequently eliminated by CAT or GPX, with CAT targeting hydrogen peroxide and GPX specializing in lipid peroxides [245]. HO-1 and NQO1 provide additional protection by promoting the breakdown of heme and the reduction in quinones, respectively [246]. The regulation of these enzymes occurs at multiple levels, with the Nrf2/Keap1 signaling pathway playing a pivotal role [247].

Numerous experimental studies have reported that various EOs and their constituents can modulate both the activity and expression of these enzymes, often via the Nrf2 axis or related signaling cascades (see Table 5 for an accurate summary). To reliably evaluate these effects, a range of analytical approaches is commonly applied. These include conventional enzyme assays for activity quantification, Western blotting and ELISA for protein measurement, immunofluorescence imaging, and transcript-level assessments by qRT-PCR. With the rise in high-throughput and integrative omics technologies, it is now possible to map enzyme networks and interactions more comprehensively, revealing the broader impact of EOs on cellular antioxidant capacity [248].

Most of the published data come from in vitro studies, raising questions about translational relevance and clinical applicability. Therefore, future work should prioritize the deconvolution of structure–activity relationships, rigorous validation in animal models or clinical settings, and the use of integrative omics to clarify the molecular landscape underlying EO-driven redox modulation.

### 6.4. Frontier Biotechnologies for Target Identification and Validation

Recent advances in chemical biology and proteomics have greatly expanded the possibilities for identifying molecular targets and elucidating the mechanisms of antioxidant compounds. Although these approaches have long driven progress in synthetic drug discovery, their broader application to EOs and phytochemicals is still emerging and is expected to reveal novel molecular mechanisms and support the rational design of next-generation antioxidant agents (Figure 7) [249,250,251]. Figure 7Integrative strategy combining computational profiling and experimental identification to elucidate the antioxidant targets of EOs. The upper panel outlines the computational workflow for predicting interactions between EO constituents and antioxidant-related targets. The lower panel shows the experimental validation used to assess antioxidant activity and target engagement, providing mechanistic insight into the antioxidant action of EOs.
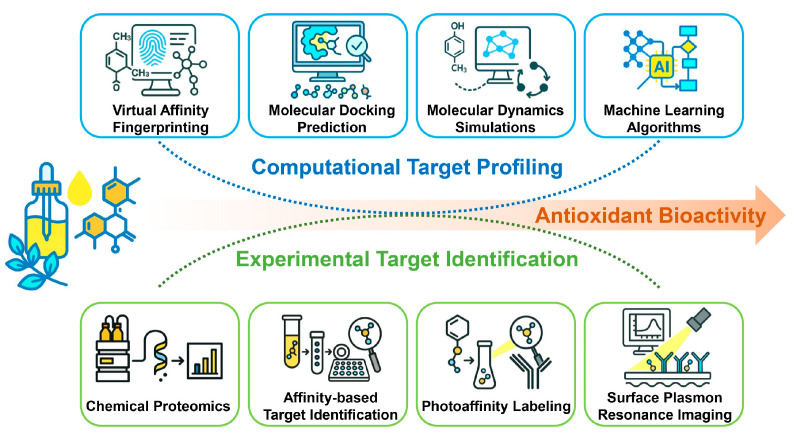


Most EO constituents are mono- and sesquiterpenes, which are small, low-polarity molecules with limited hydrogen-bonding capacity. Their volatility, hydrophobicity, and relatively weak binding affinities make direct biophysical characterization, such as X-ray crystallography, surface plasmon resonance (SPR), or isothermal titration calorimetry (ITC), technically challenging [167]. Nevertheless, these terpenoids appear to interact with protein targets in a selective rather than random manner. Specific structural motifs such as hydroxyl, carbonyl, double bond, or epoxide groups (in oxidized EO components) may explain target recognition and biological activity. Previous studies combining radioligand binding, electrophysiology, and selective antagonism have experimentally verified defined interactions between typical EO components including β-caryophyllene, eugenol, linalool, menthol, carvacrol, and thymol, and receptor or ion-channel targets such as cannabinoid receptor type 2 (CB_2_), transient receptor potential ankyrin 1 (TRPA1), N-methyl-D-aspartate receptor (NMDA receptor), and γ-aminobutyric acid type A receptor (GABA_A_) [252,253,254,255,256].

Comprehensive chemoproteomic strategies such as photoaffinity labeling (PAL) coupled with high-resolution mass spectrometry have proven highly effective for identifying covalent or transient binding partners under near-physiological conditions [167,250]. When combined with computational docking and molecular dynamics simulations, these tools can distinguish specific protein interactions from nonspecific effects and provide mechanistic insights into the biological activity of EO-derived compounds [257,258]. For early-stage screening or phenomenological studies, however, conventional biochemical and cell-based assays generally provide sufficient information without requiring large-scale proteomic analysis. A tiered workflow that begins with in vitro and in silico screening, followed by chemoproteomic validation only when mechanistic clarification is necessary, offers a balanced and cost-effective approach for EO research.

## 7. Conclusions and Perspectives

EOs represent a structurally and functionally diverse group of natural antioxidants whose actions extend beyond classical radical-scavenging chemistry to encompass complex cellular redox regulation. Current evidence indicates that EO constituents can act through multiple layers of antioxidant defense, ranging from chain-breaking reactions and lipid peroxidation inhibition to the modulation of redox-sensitive signaling pathways. Phenolic components, together with compounds containing electrophilic α,β-unsaturated aldehyde structures, can directly activate the Nrf2/Keap1 axis through covalent modification of Keap1 cysteine residues, whereas non-phenolic terpenes may exert similar effects following oxidative metabolism or through indirect modulation of upstream kinases. Despite extensive biochemical and cellular data, definitive biophysical evidence confirming the direct interaction between EO-derived molecules and their target proteins remains limited. Future studies should integrate chemical reactivity profiling, kinetic modeling, and advanced bioanalytical techniques to verify specific binding events and clarify structure–activity relationships. A deeper mechanistic understanding of how EO constituents influence redox homeostasis will not only reinforce their scientific validation but also facilitate their rational application as natural preservatives, redox modulators, and health-promoting agents in food and pharmacological systems.

Future investigations of EOs as redox-active systems are expected to shift from descriptive antioxidant screening toward mechanism-driven and quantitatively controlled experimental designs. Greater reliance on lipid peroxidation kinetics, oxygen-consumption measurements, high-resolution lipidomics, and compartment-specific ROS imaging will be essential for elucidating how individual EO constituents and their combinations influence chain propagation and termination within membrane-associated oxidative processes. At the biological level, establishing causal links between lipid peroxidation control, Nrf2/Keap1-centered redox signaling, and disease-relevant cellular phenotypes represents a critical next step. In this context, diseases in which dysregulated lipid peroxidation acts as a central pathogenic driver provide relevant biological frameworks to explore whether EO-mediated regulation of membrane lipid oxidation and associated redox pathways can contribute to disease-relevant protective phenotypes.

The intrinsic chemical instability, compositional variability, and multicomponent nature of EOs continue to pose major challenges for reproducible mechanistic research and translational interpretation. The lack of harmonized standards for chemical characterization, storage, and stability assessment remains a major source of inter-study variability and uncertainty in structure–function relationships. The implementation of standardized quality-control frameworks, supported by periodic GC–MS verification, will be indispensable for distinguishing native bioactivity from storage-induced artifacts and secondary oxidation effects. Given the complexity of EO mixtures, data-driven strategies such as multivariate modeling, machine learning, network pharmacology, and systems redox biology will play an increasingly important role in disentangling component interactions, identifying dominant bioactive nodes, and constructing predictive structure–function models. Such integrative approaches will be pivotal for advancing EO research from empirical observation toward mechanism-guided application across food, biomedical, and preventive health contexts.

## Data Availability

No new data were created or analyzed in this study. Data sharing is not applicable to this article.

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
