# Peer review of "Essential Oils as Antioxidants: Mechanistic Insights from Radical Scavenging to Redox Signaling"

_antioxidants, 2025, doi:10.3390/antiox15010037_

Round 1
Reviewer 1 Report
The review article “Essential Oils as Antioxidants: Mechanistic Insights from Radical Scavenging to Redox Signaling” by Foti et al. discusses the antioxidant activity of essential oils and their constituents. The article provides a comprehensive overview that includes general information about essential oils and their applications, fundamental concepts related to lipid peroxidation and its products, as well as information on antioxidants and their general classification (preventive or chain reaction‐interrupting). The authors also focus on describing the antioxidant activity of both phenolic and non-phenolic components of essential oils, with particular emphasis on γ-terpinene.
A substantial portion of the article is devoted to chemical and biological methods used to evaluate the antioxidant activity of essential oils and their components. The material is presented in an engaging manner and supported by numerous figures, schemes, and tables.
In my opinion, the article can be published in the journal, but it requires minor revisions:
- The introduction should clearly and concisely state the aim of the manuscript. What exactly is the subject of the review, and what purpose does the presented content serve? Additionally, the authors should include a description of the review methodology—how the references were selected, what criteria were considered (e.g., keywords), the time frame covered (years), and the databases used (Scopus, Elsevier, PubMed, etc.).
- The authors should consider dividing some of the larger diagrams into smaller parts. For example, Scheme 1 could be split into three smaller schemes, each placed near the corresponding section in which it is discussed. In my opinion, this would improve the readability of the manuscript.
- Some figures and schemes have captions that are too lengthy, e.g., Figure 1, Scheme 3, Figures 5 and 7. In my view, part of this information should be moved to the main text.
- Schemes 2, 3, and 4 lack information about how they were created; their descriptions contain no references. The same issue applies to some figures (e.g., Figure 5).
- Subsection 4.3: Why do the authors focus exclusively on the DPPH, ABTS, and FRAP methods? Other radical-based methods are also available.
- Tables 1 and 2 are not introduced or summarized in the text. Additionally, in line 593, the reference should be to “Table 4” rather than “Table 5.”
- All abbreviations should be defined upon first use throughout the manuscript.
- Table 3: The authors should add a column providing a proposed explanation for the antioxidant activity of each essential oil or component.
- Table 4: The column titled “Inducer” should be combined with, or placed closer to, the column titled “Induced Models.”
- All figures and schemes should be approximately the same size.
The review article “Essential Oils as Antioxidants: Mechanistic Insights from Radical Scavenging to Redox Signaling” by Foti et al. discusses the antioxidant activity of essential oils and their constituents. The article provides a comprehensive overview that includes general information about essential oils and their applications, fundamental concepts related to lipid peroxidation and its products, as well as information on antioxidants and their general classification (preventive or chain reaction‐interrupting). The authors also focus on describing the antioxidant activity of both phenolic and non-phenolic components of essential oils, with particular emphasis on γ-terpinene.
A substantial portion of the article is devoted to chemical and biological methods used to evaluate the antioxidant activity of essential oils and their components. The material is presented in an engaging manner and supported by numerous figures, schemes, and tables.
In my opinion, the article can be published in the journal, but it requires minor revisions:
- The introduction should clearly and concisely state the aim of the manuscript. What exactly is the subject of the review, and what purpose does the presented content serve? Additionally, the authors should include a description of the review methodology—how the references were selected, what criteria were considered (e.g., keywords), the time frame covered (years), and the databases used (Scopus, Elsevier, PubMed, etc.).
- The authors should consider dividing some of the larger diagrams into smaller parts. For example, Scheme 1 could be split into three smaller schemes, each placed near the corresponding section in which it is discussed. In my opinion, this would improve the readability of the manuscript.
- Some figures and schemes have captions that are too lengthy, e.g., Figure 1, Scheme 3, Figures 5 and 7. In my view, part of this information should be moved to the main text.
- Schemes 2, 3, and 4 lack information about how they were created; their descriptions contain no references. The same issue applies to some figures (e.g., Figure 5).
- Subsection 4.3: Why do the authors focus exclusively on the DPPH, ABTS, and FRAP methods? Other radical-based methods are also available.
- Tables 1 and 2 are not introduced or summarized in the text. Additionally, in line 593, the reference should be to “Table 4” rather than “Table 5.”
- All abbreviations should be defined upon first use throughout the manuscript.
- Table 3: The authors should add a column providing a proposed explanation for the antioxidant activity of each essential oil or component.
- Table 4: The column titled “Inducer” should be combined with, or placed closer to, the column titled “Induced Models.”
- All figures and schemes should be approximately the same size.
Author Response
The review article “Essential Oils as Antioxidants: Mechanistic Insights from Radical Scavenging to Redox Signaling” by Foti et al. discusses the antioxidant activity of essential oils and their constituents. The article provides a comprehensive overview that includes general information about essential oils and their applications, fundamental concepts related to lipid peroxidation and its products, as well as information on antioxidants and their general classification (preventive or chain reaction‐interrupting). The authors also focus on describing the antioxidant activity of both phenolic and non-phenolic components of essential oils, with particular emphasis on γ-terpinene.
A substantial portion of the article is devoted to chemical and biological methods used to evaluate the antioxidant activity of essential oils and their components. The material is presented in an engaging manner and supported by numerous figures, schemes, and tables.
In my opinion, the article can be published in the journal, but it requires minor revisions:
- The introduction should clearly and concisely state the aim of the manuscript. What exactly is the subject of the review, and what purpose does the presented content serve? Additionally, the authors should include a description of the review methodology—how the references were selected, what criteria were considered (e.g., keywords), the time frame covered (years), and the databases used (Scopus, Elsevier, PubMed, etc.).
Response: We thank the reviewer for this important comment. The Introduction has been revised to define the scope and objectives of the review (lines 114-124). In addition, a description of the review methodology, including literature selection criteria, keywords, time frame, and databases used, has now been included in the revised manuscript (lines 125-135).
- The authors should consider dividing some of the larger diagrams into smaller parts. For example, Scheme 1 could be split into three smaller schemes, each placed near the corresponding section in which it is discussed. In my opinion, this would improve the readability of the manuscript.
Response: We appreciate this suggestion. The relevant schemes have been reorganized and divided into smaller parts, which are now placed closer to the corresponding sections in the text to improve readability.
- Some figures and schemes have captions that are too lengthy, e.g., Figure 1, Scheme 3, Figures 5 and 7. In my view, part of this information should be moved to the main text.
Response: We agree with the reviewer. The captions of the indicated figures and schemes have been shortened, and part of the descriptive information has been moved to the main text.
- Schemes 2, 3, and 4 lack information about how they were created; their descriptions contain no references. The same issue applies to some figures (e.g., Figure 5).
Response: We thank the reviewer for raising this point. The primary literature sources have been explicitly indicated in the caption.
- Subsection 4.3: Why do the authors focus exclusively on the DPPH, ABTS, and FRAP methods? Other radical-based methods are also available.
Response: We appreciate this comment. To broaden the methodological perspective, the discussion has been expanded to include the aqueous-phase ORAC assay and the lipid-relevant FENIX approach, which are now discussed in Lines 524–556 of the revised manuscript.
- Tables 1 and 2 are not introduced or summarized in the text. Additionally, in line 593, the reference should be to “Table 4” rather than “Table 5.”
Response: We agree with the reviewer. Tables 1 and 2 are now explicitly introduced and summarized in the main text, and the incorrect table citation has been corrected.
- All abbreviations should be defined upon first use throughout the manuscript.
Response: This has been corrected throughout the manuscript.
- Table 3: The authors should add a column providing a proposed explanation for the antioxidant activity of each essential oil or component.
Response: We thank the reviewer for this helpful suggestion. An additional column providing a proposed mechanistic explanation for the antioxidant activity has been added to Table 3.
- Table 4: The column titled “Inducer” should be combined with, or placed closer to, the column titled “Induced Models.”
Response: The table has been reorganized accordingly in the revised manuscript.
- All figures and schemes should be approximately the same size.
Response: The sizes of all figures and schemes have been adjusted to improve visual consistency.
Reviewer 2 Report
In the manuscript, the authors reviewed essential oils as antioxidants, focusing on mechanistic insights ranging from radical scavenging to redox signaling. Overall, it is a well-written review. However, I have a few comments for improvement:
1. Given the primary emphasis on lipid peroxidation, it would be beneficial for readers if the authors also provided a broader discussion on ferroptosis. Currently, the manuscript includes only limited information on this topic.
2. It would be helpful for the authors to include details about the stability of essential oils.
3. Since mitochondria are the primary organelles involved in redox regulation, a more comprehensive discussion on mitochondria would enhance the reader's understanding.
4. The MDA assay should be discussed, as it is widely used to identify the lipid peroxidation inhibitory activity of antioxidants.
5. Including a schematic representation of a colorimetric assay, such as the ABTS assay, like Scheme 5, would be beneficial for readers to follow along.
6. Is there a way to distinguish between the RTA effect and the intracellular effect of essential oils when assessing their antioxidant effects in a specific model?
NA
Author Response
In the manuscript, the authors reviewed essential oils as antioxidants, focusing on mechanistic insights ranging from radical scavenging to redox signaling. Overall, it is a well-written review. However, I have a few comments for improvement:
- Given the primary emphasis on lipid peroxidation, it would be beneficial for readers if the authors also provided a broader discussion on ferroptosis. Currently, the manuscript includes only limited information on this topic.
Response: We thank the reviewer for this valuable suggestion. We agree that ferroptosis is closely linked to lipid peroxidation and is of high relevance to this field. A concise description of this relationship, together with several representative examples involving essential oils, has already been included in Lines 979–984. At present, however, systematic studies on ferroptosis regulation by essential oil components remain very limited, and no integrated mechanistic network has yet been established. Most advances in this area concern drugs or non-volatile natural products and are not directly applicable to essential oils. Nevertheless, following the reviewer’s suggestion, we have added further perspective and discussion on the potential integration of ferroptosis into future essential oil antioxidant research.
- It would be helpful for the authors to include details about the stability of essential oils.
Response: A discussion on the stability of essential oils has now been added to the manuscript (Lines 468–476).
- Since mitochondria are the primary organelles involved in redox regulation, a more comprehensive discussion on mitochondria would enhance the reader's understanding.
Response: We appreciate this comment. The role of mitochondria in cellular redox regulation has been expanded and discussed in the revised manuscript (Lines 605–613).
- The MDA assay should be discussed, as it is widely used to identify the lipid peroxidation inhibitory activity of antioxidants.
Response: Methods for MDA determination, including thiobarbituric acid–based assays and HPLC-based approaches, have been added (Lines 678–691).
- Including a schematic representation of a colorimetric assay, such as the ABTS assay, like Scheme 5, would be beneficial for readers to follow along.
Response: We thank the reviewer for this helpful suggestion. We agree that schematic representations of colorimetric assays, such as the ABTS assay, can be useful for instructional purposes. However, given that such assays are already extensively illustrated in the existing literature and numerous previous reviews, we have chosen not to include an additional schematic in order to avoid redundancy and to maintain the focus of the present review on mechanistic and methodological aspects that are less comprehensively covered elsewhere.
- Is there a way to distinguish between the RTA effect and the intracellular effect of essential oils when assessing their antioxidant effects in a specific model?
Response: We thank the reviewer for this insightful question. In principle, a distinction between radical-trapping antioxidant (RTA) effects and intracellular, pathway-mediated antioxidant responses can be made when independent readouts of chemical radical-trapping activity and cellular signaling or enzymatic induction are available. Direct RTA activity is expected to afford protection without the activation of NRF2 or related redox-regulated pathways, whereas compounds lacking intrinsic radical-trapping capability, such as unsaturated aldehydes, are more likely to exert protective effects primarily through the induction of endogenous antioxidant defenses.
In practice, however, this distinction is often blurred, as RTA metabolites can be electrophilic and may also trigger NRF2 signaling. An illustrative example is provided by our recent study (Z. Jin et al., Chem. Eur. J., 2024), in which γ-terpinene protected membrane lipids and suppressed ferroptotic cell death without detectable NRF2 activation, consistent with a predominantly direct, chain-breaking RTA mechanism in that specific model.
Reviewer 3 Report
I do not have major comments.
Dear authors,
I have reviewed the paper and I believe that this is a comprehensive review paper that describes in detail the reactive oxygen species from their formation to decomposition, as well as the influence of essential oils on this process.
I have only a few minor suggestions.
Comment 1:
Please add section about novels and future directions/suggestions.
Comment 2:
Please add section about apoptotic pathway and how oxidative stress regulates apoptosis, and how essential oils may modulate it: Bax/Bcl-2 ratio, Caspase-3, -9 activation, Cytochrome c release, JNK/p38 MAPK activation.
Author Response
Dear authors,
I have reviewed the paper and I believe that this is a comprehensive review paper that describes in detail the reactive oxygen species from their formation to decomposition, as well as the influence of essential oils on this process.
I have only a few minor suggestions.
Comment 1:
Please add section about novels and future directions/suggestions.
Response: We thank the reviewer for this helpful suggestion. Following the reviewer’s recommendation, we have expanded the Conclusion section to a new “Conclusion and Perspectives” section, in which we explicitly discuss the novelty of the present review and outline future directions and methodological perspectives for essential oil antioxidant research.
Comment 2:
Please add section about apoptotic pathway and how oxidative stress regulates apoptosis, and how essential oils may modulate it: Bax/Bcl-2 ratio, Caspase-3, -9 activation, Cytochrome c release, JNK/p38 MAPK activation.
Response: Thank you for this valuable suggestion. We fully agree that oxidative stress is deeply involved in the regulation of apoptosis, through mechanisms such as modulation of the Bax/Bcl-2 ratio, activation of caspase-9 and caspase-3, mitochondrial cytochrome c release, and stress-activated JNK and p38 MAPK signalling. These redox-sensitive pathways are undoubtedly important in the broader context of cell fate regulation.
However, the primary aim of the present review is to provide a focused mechanistic framework for intracellular antioxidant defence, with particular emphasis on redox homeostasis and ferroptosis-related pathways, especially the Nrf2–Keap1 axis. Apoptotic signalling, although closely connected to oxidative stress, involves a distinct regulatory logic and extensive upstream and downstream networks that fall outside the central scope of this work. For this reason, we have chosen not to expand the manuscript into apoptosis-related mechanisms.